# Box C/D small nucleolar ribonucleoproteins regulate mitochondrial surveillance and innate immunity

Elissa Tjahjono, Alexey V. Revtovich, Natalia V. Kirienko *

Department of BioSciences, Rice University, Houston, Texas, United States of America

* kirienko@rice.edu

## Abstract

Monitoring mitochondrial function is crucial for organismal survival. This task is performed by mitochondrial surveillance or quality control pathways, which are activated by signals originating from mitochondria and relayed to the nucleus (retrograde response) to start transcription of protective genes. In *Caenorhabditis elegans*, several systems are known to play this role, including the UPR$^{mt}$, MAPK$^{mt}$, and the ESRE pathways. These pathways are highly conserved and their loss compromises survival following mitochondrial stress. In this study, we found a novel interaction between the box C/D snoRNA core proteins (snoRNPs) and mitochondrial surveillance and innate immune pathways. We showed that box C/D, but not box H/ACA, snoRNPs are required for the full function of UPR$^{mt}$ and ESRE upon stress. The loss of box C/D snoRNPs reduced mitochondrial mass, mitochondrial membrane potential, and oxygen consumption rate, indicating overall degradation of mitochondrial function. Concomitantly, the loss of C/D snoRNPs increased immune response and reduced host intestinal colonization by infectious bacteria, improving host resistance to pathogenesis. Our data may indicate a model wherein box C/D snoRNP machinery regulates a "switch" of the cell's activity between mitochondrial surveillance and innate immune activation. Understanding this mechanism is likely to be important for understanding multifactorial processes, including responses to infection and aging.

## Author summary

Mitochondrial health is important for organismal survival. Multiple cellular pathways are dedicated to actively monitoring mitochondrial status, termed as the mitochondrial surveillance system, to provide better defense towards variety of stresses. These systems are highly conserved and present in both *C. elegans* and humans. Here we report that the Box C/D snoRNA core proteins (snoRNPs), normally associated with modification of ribosomal RNA, play a role in mitochondrial surveillance and innate immune pathways. The loss of this protein complex reduced mitochondrial surveillance pathway activation after stress but increased immune responses. As mitochondrial surveillance, innate immunity, and box C/D snoRNP pathways are conserved in humans, understanding their roles

**Data Availability Statement:** All relevant data are within the manuscript and its Supporting Information files.

**Funding:** NVK, a CPRIT scholar in Cancer Research, thanks the Cancer Prevention and

Research Institute of Texas (CPRIT, https://www.
cprit.state.tx.us/) for their generous support, CPRIT
grant RR150044. This work was also supported by
the National Institutes of Health (NIGMS
R35GM129294, https://www.nigms.nih.gov/) to
NVK. Funders play no role in the study design, data
collection and analysis, decision to publish, or
preparation of the manuscript.

**Competing interests:** The authors have declared
that no competing interests exist.

in modulating cell biology during mitochondrial crises is essential for mitigating stress
and restoring health after damage.

## Introduction

All living organisms require the maintenance of cellular homeostasis under conditions very
different than their surroundings. Maintaining these conditions requires constant surveillance
for disruption and metabolic adjustments to reacquire the proper biochemical balance. Mean-
while, a variety of insults can disrupt this balance, ranging from environmental changes to
metabolic dysfunction to pathogen infection. Indeed, almost all cellular pathways are dis-
rupted in one infection or another, including protein translation [1,2], the proteostatic
machinery [3–5], the cytoskeleton [6], the endoplasmic reticulum [7,8], and others [9,10].

Given the central role of mitochondria in energy production, biosynthesis of heme groups,
lipid metabolism, the regulation of iron and calcium homeostasis, and production of reactive
oxygen species (ROS), it should be no surprise that mitochondria are impacted by disease and
infection [11–13]. Consequently, they are subjected to several important surveillance path-
ways. The two best known are the PINK1/Parkin axis for macroautophagic mitochondrial
recycling (commonly known as mitophagy) and the unfolded protein response in mitochon-
dria (UPR$^{mt}$) [14–17]. Both systems monitor the functionality of mitochondrial protein import
and they have been thoroughly investigated in various model organisms, including *C. elegans*,
yeast, and mice. For example, in the former pathway, accumulation of the kinase PINK1 on
the outer mitochondrial surface licenses its ability to phosphorylate its targets, including Par-
kin, resulting in the subsequent recruitment of autophagic machinery. Under similar condi-
tions of compromised mitochondrial import, the key transcription factor ATFS-1/ATF5 is
redirected from mitochondria, where it would be degraded, to the nucleus, where it regulates
the expression of chaperones and other stress mediators.

A third, rather more elusive, mitochondrial pathway utilizes the DLK-1/SEK-3/PMK-3
MAP kinase cascade (which we will refer to as the MAPK$^{mt}$ pathway) and was identified in *C.
elegans* by activating mitochondrial stress and searching for differentially expressed genes that
were independent of ATFS-1/ATF5 [18]. Regulation of this pathway appears to involve a C/
EBP family transcription factor called CBP-3 and disrupted mitochondrial electron transport
chain (ETC) function. The MAPK$^{mt}$ pathway is involved in the extended lifespan observed in
long-lived mitochondrial (Mit) mutants [18]. Interestingly, fluvastatin, which disrupts mevalo-
nate metabolism and prevents geranylgeranylation of certain components of the vesicular traf-
ficking system, also activates the MAPK$^{mt}$ pathway, indicating that surveillance of
mitochondrial cholesterol metabolism is also important [19].

Our lab has previously identified a key mitochondrial surveillance program in *C. elegans*
regulated by cellular ROS [20,21]. This pathway, known as the Ethanol and Stress Response
(ESRE) network, is named for an 11-nucleotide motif found in the promoter region of hun-
dreds of genes in *C. elegans* and ethanol-responsive genes in mice, and is activated by a range
of abiotic triggers [22–24]. Interestingly, exposure to the opportunistic human pathogen, *Pseu-
domonas aeruginosa*, which produces a xenobiotic siderophore called pyoverdine that hijacks
mitochondria-resident iron from *C. elegans*, also activates the ESRE network [20,25,26]. This
also appears to be the first known stress pathway to respond to intracellular reductive stress
[21].

Active study from our lab and others has linked several determinants to ESRE gene expres-
sion, including the JumonjiC-domain containing protein JMJC-1/Riox1 (also known as

NO66) [22], the PBAF nucleosome remodeling complex [27], a family of bZIP transcription factors (ZIP-2, ZIP-4, CEBP-1, and CEBP-2) [20], and the Zn-finger transcription factor SLR-2 [22,28]. Importantly, the ESRE motif, the genes regulated by it, and their activation in response to stress are ancient and evolutionarily conserved from *C. elegans* to humans [22,24].

Mitochondrial surveillance programs not only activate programs to reacquire homeostasis, they also regulate innate immune functions [25,29–32], a process sometimes termed surveillance immunity [33]. However, innate immune activation is energetically costly and requires considerable energy conversion [34,35] and excess immune activity is associated with a broad range of deleterious health outcomes. Thus, it behooves the organism to carefully balance its need to reacquire homeostasis and repair damage against stimulating innate immune functions that can cause further damage. How organisms navigate this choice remains a poorly understood area of biology.

Small nucleolar ribonucleoproteins (snoRNPs) are small complexes of RNA and protein best known for catalyzing site-directed modifications to RNA. snoRNPs fall into two major groups, called the box C/D and box H/ACA families, that are categorized based on their functions and the secondary structures of their snoRNA components [36]. Box C/D snoRNPs, consisting of FIB-1/ Fibrillarin (the catalytic methyltransferase), NOL-56/Nop56, NOL-58/Nop58, and M28.5/SNU13 (**Fig 1A, left**), perform 2'-O-methylation, while box H/ACA snoRNPs, comprised of Y66H1A.4/ Gar1, Y48A6B.3/Nhp2, NOLA-3/Nop10, and K01G5.5/dyskerin, convert targeted uridine residues to pseudouridine (**Fig 1A, right**) [37]. Both families mainly target ribosomal RNA, with the modifications typically clustered at biologically important locations [38]. The snoRNA in each complex confers the targeting function via sequence complementarity to the modification target site; it helps to orient and target the enzymatic function of the snoRNP complex [38]. snoRNP complexes have been suggested to have functional roles well beyond the processing of rRNA, including 2'-O-methylation, splicing, and translation of mRNAs [39–43].

In this study, we identified a non-canonical role for the box C/D snoRNP complex, which appears to serve as a molecular switch that activates mitochondrial surveillance and represses conventional innate immune processes. Specifically, box C/D snoRNPs upregulate ESRE and UPR[mt] while downregulating the function of the PMK-1/p38 MAPK pathway. Contrarily, knockdown of box C/D snoRNPs upregulated MAPK[mt] pathway effectors, but this was likely a secondary effect from the loss of MAPK[mt] repression by the UPR[mt], which is characteristic of the complicated interactions between these surveillance systems. Since box C/D snoRNPs and these mitochondrial surveillance systems are conserved between *C. elegans* and humans, our results may lead to a better understanding of processes that affect mitochondrial health and innate immune pathways in human diseases.

## Results

### Identification of FIB-1/Fibrillarin and NOL-56/Nop56 as regulators of the ESRE pathway

To identify additional regulatory components of the ESRE pathway, a biotinylated oligonucleotide comprised of a 4x tandem repeat of the consensus 11-nucleotide ESRE motif was used as bait for biochemical pulldown. Young adult *C. elegans* were exposed to either DMSO, 1 mM phenanthroline (a chemical iron chelator), or 50 μM rotenone (an inhibitor of ETC Complex I) to trigger mitochondrial damage and ESRE gene activation [20]. Proteins were extracted from the cytoplasm and nuclei and mixed with the biotinylated ESRE bait and then pulled down using streptavidin-coated magnetic beads. An electrophoretic mobility shift assay (EMSA) [27] was used to optimize enrichment of ESRE-binding proteins and to verify specificity (**Fig 1B**).

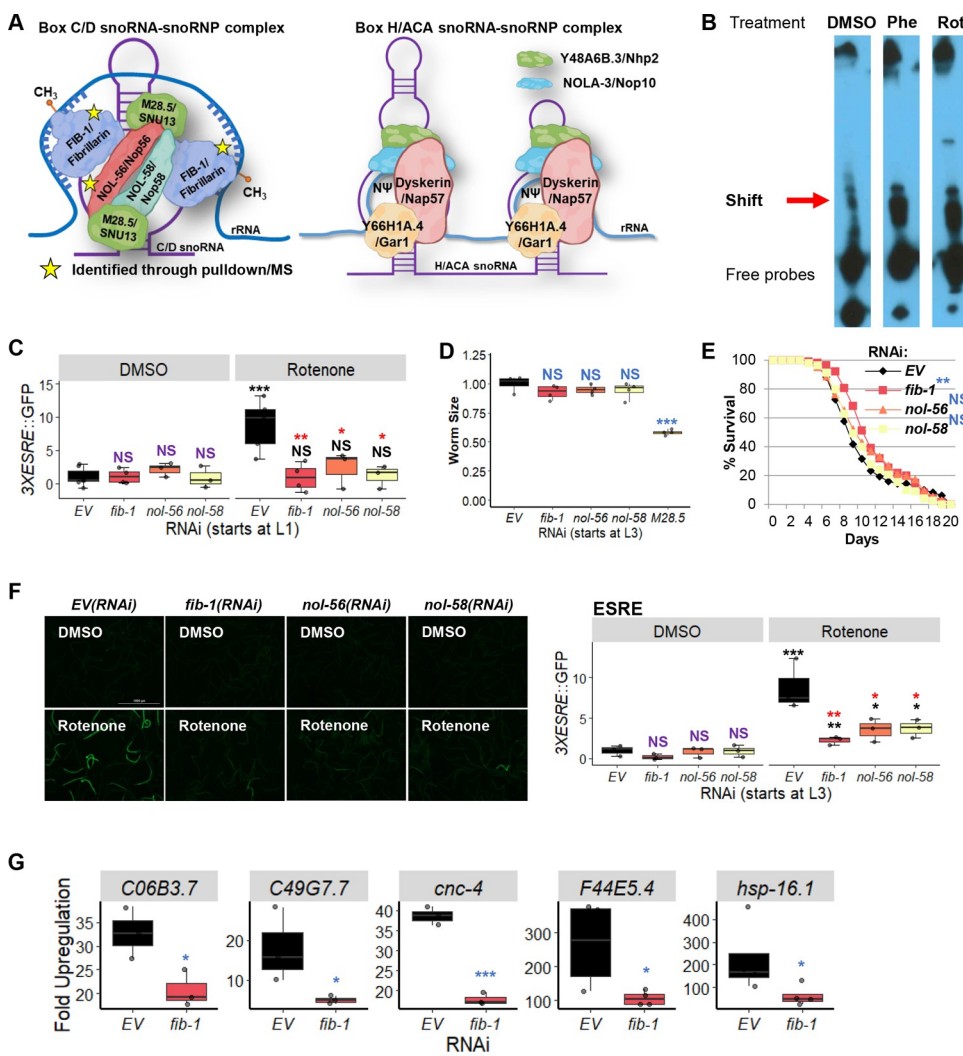

**Fig 1. Box C/D snoRNP machinery binds to the ESRE motif.** **(A)** Cartoon representation of box C/D (*left*) or box H/ACA (*right*) snoRNA-snoRNP complexes. Stars indicate proteins identified through oligo pulldown-mass spectrometry. **(B)** Electrophoretic mobility shift assay (EMSA) showed the presence of the ESRE-binding motif through the identified 'Shift'. **(C)** Quantification of GFP fluorescence of *C. elegans* carrying a *3XESRE*::GFP reporter that was reared on *E. coli* expressing RNAi targeting empty vector (*EV*) or *fib-1(RNAi)*, *nol-56(RNAi)*, or *nol-58 (RNAi)*. **(D)** The size and **(E)** lifespan of *glp-4(bn2)* worms grown on RNAi strains targeting Box C/D snoRNPs are shown. **(F)** Fluorescent images (*left*) and quantification of GFP fluorescence (*right*) of *C. elegans* carrying a *3XESRE*::GFP reporter reared on *E. coli* expressing RNAi targeting empty vector (*EV*), *fib-1(RNAi)*, *nol-56(RNAi)*, or *nol-58 (RNAi)*. **(G)** Upregulation of ESRE gene transcripts as measured via qRT-PCR. Worms were reared on *E. coli* expressing RNAi targeting empty vector *(EV)* or *fib-1(RNAi)* and were treated with DMSO or 25 μM rotenone for 8 hours. RNAi treatment was started at the L1 **(C)** or L3 **(D-G)** stage. Representative images are shown. Three biological replicates comprised of ~400 **(C, D, F)**, ~150 **(E)**, or ~8,000 **(G)** worms/replicate were analyzed. A representative replicate for the lifespan assay is shown **(E)**. **(C, F)** Worms were treated for 8 hours with vehicle (DMSO) or 50 μM rotenone and GFP values were normalized to *EV*-DMSO. Worms' sizes were normalized to *EV* control. *p* values were determined from two-way or one-way ANOVA for ESRE expression and worms' size measurement, respectively, followed by Dunnett's test, log-rank test for lifespan assay, or Student's *t*-test for qRT-PCR. NS not significant, * $p < 0.05$, ** $p < 0.01$, *** $p < 0.001$. In panels with colored significance marks, purple indicates comparison between *gene(RNAi)* and *EV(RNAi)* in control condition (DMSO), red indicates comparison between *gene(RNAi)* and *EV (RNAi)* in stressed condition (rotenone), and black indicates comparison between stressed and control conditions.

We detected multiple bands from the EMSA, labeled as 'Shift' and 'Free probes' (Fig 1B). The 'Free probes' are excess biotinylated ESRE oligos that did not bind protein. Multiple 'Shift' bands were observed, which may indicate constitutive binding of the ESRE motif or a

different activated form of the transcription factor [44]. These bands were thicker upon treatment with phenanthroline or rotenone, indicating an increase in DNA-binding activity (**Fig 1B**). An EMSA experiment was also performed using mutated ESRE sites (three mutations per element) to confirm specificity. These substitutions considerably reduced binding by material from rotenone-treated worms compared to wild-type ESRE (**S1 Fig**). These results confirmed that one or more stress-inducible nuclear factor(s) bind the ESRE motif after mitochondrial damage and suggested that it may be required for the transcription of the ESRE genes. Materials bound under these conditions were eluted and subjected to tandem MS/MS analysis for identification of potential peptide fragments.

The gene products identified by mass spectrometry were categorized by using the 'SRA' binning system and the 'iBAQ' score. The 'SRA' or 'Strict, Relaxed, and All' binning approach utilizes tiered metrics to score gene identification quality, in which the identified 'Strict' genes products pass a 1% FDR cutoff [45]. Meanwhile, the 'iBAQ' scores were calculated based on peptide peak intensities and number of potential peptides, comparable to the absolute protein quantity. By searching for proteins that passed the 'SRA' binning system as 'Strict' and that are enriched in rotenone-treated samples compared to DMSO control, 75 candidate proteins were identified (**S1 Table**).

To establish a role in ESRE function, each gene predicted to encode one of these proteins was knocked down via RNAi in a strain of *C. elegans* carrying an ESRE-GFP transcriptional reporter (3x tandem repeat of the ESRE consensus sequence driving a GFP reporter, *3xESRE*, GFP) [21]. After knockdown, activation of the reporter was induced using 50 μM rotenone. Amongst the candidate genes, only RNAi targeting *fib-1/Fibrillarin* and *nol-56/Nop56* reduced reporter expression (**Fig 1C**). As NOL-58 is a third member of the box C/D snoRNP complex (**Fig 1A**) but was not identified in the affinity-purified material, *nol-58/Nop58(RNAi)* was tested for the ability to reduce ESRE-GFP expression. Like its other box C/D snoRNP counterparts, RNAi targeting *nol-58/Nop58* reduced ESRE expression (**Fig 1C**). The current model of box C/D snoRNP assembly suggests that FIB-1/Fibrillarin and NOL-58/Nop58 independently bind to the snoRNA and then NOL-56/Nop56 associates with the complex, but does not bind to the snoRNA alone [46]. Since knockdown of any of the genes for these three proteins reduced ESRE signaling, it seems likely that the snoRNP complex as a whole is binding to ESRE.

Although worms reared on RNAi targeting *fib-1/FBL*, *nol-56/Nop56*, or *nol-58/Nop58* exhibited reductions in ESRE signaling, they also showed clear signs of reduced growth and development, giving rise to smaller adults, which is consistent with a previous report [47]. To reduce the possibility that our observations were due to non-specific developmental effects, we tested the impact of disrupting snoRNP targets after the L3 stage. Beginning RNAi at this developmental stage eliminated overt effects on final length of the worms (**Fig 1D**), except when *M28.5(RNAi)* was used. As such, *M28.5(RNAi)* was excluded from further experiments. Starting RNAi at L3 not only mitigated the negative impact that snoRNP RNAi had on *C. elegans* lifespan (**Fig 1E** and **S2 Table**), but also extended lifespan for *fib-1(RNAi)* (similar to what was observed in [41]), suggesting overall increased health.

Exposure at a later stage of development can circumvent some of the developmental effects and apparent general malaise, but it can also reduce the penetrance of the RNAi phenotype [48]. In this case, however, snoRNP knockdown at L3 stage still reduced ESRE activation upon stress (**Fig 1F**). We also quantified gene transcripts in worms reared on *E. coli* expressing empty vector (*EV*) or *fib-1(RNAi)*. Expression of ESRE genes in *fib-1(RNAi)* worms in response to rotenone was reduced compared to controls (**Fig 1G**). These data confirm that regulation of ESRE pathway by box C/D snoRNPs is specific and differs from their requirement during early developmental stages for overall *C. elegans* health.

## Box C/D snoRNPs also regulate UPR^mt and MAPK^mt

To assess whether knocking down box C/D snoRNPs only affected the ESRE pathway or also impacted other mitochondrial stress responses, we triggered activation of downstream effectors for UPR$^{mt}$ and MAPK$^{mt}$ using RNAi targeting the mitochondria-resident protease SPG-7/ AFG3L2 (*spg-7/SPG7(RNAi)*), which efficiently induces UPR$^{mt}$ [18,29,49]. Young adult worms carrying GFP reporters for the UPR$^{mt}$ (*Phsp-6*::GFP) or MAPK$^{mt}$ (*Ptbb-6*::GFP) surveillance pathways were reared on plates containing all pairwise mixtures of RNAi: empty vector or *spg-7/SPG7(RNAi)* with empty vector, *fib-1(RNAi)*, *nol-56(RNAi)* or *nol-58(RNAi)*. As with the ESRE pathway, knockdown of *fib-1*, *nol-56*, or *nol-58* reduced the ability of the UPR$^{mt}$ to respond to stress, regardless of whether RNAi was begun at the L1 or L3 stage (**Fig 2A and 2B**).

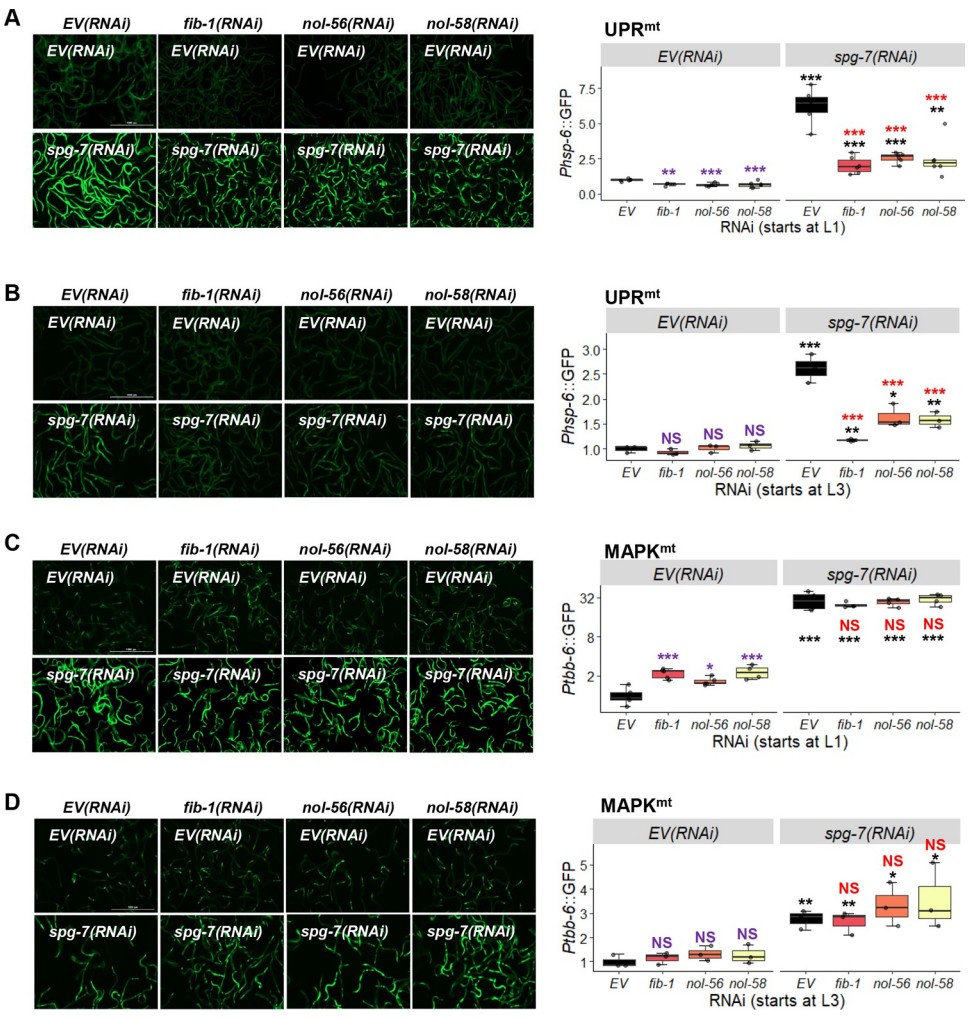

**Fig 2. Box C/D snoRNP knockdown reduces UPR^mt but increased MAPK^mt expression. (A-D)** Fluorescent images (left) and quantification of GFP fluorescence (right) of *C. elegans* carrying *Phsp-6*::GFP **(A, B)** or *Ptbb-6*::GFP **(C, D)** reporters reared on *E. coli* expressing empty vector (*EV*), *fib-1(RNAi)*, *nol-56(RNAi)*, or *nol-58(RNAi)*. Each RNAi construct was paired with either empty vector (*EV*) or *spg-7(RNAi)*. RNAi treatment was started at L1 **(A, C)** or L3 **(B, D)** stage. Representative images are shown; three biological replicates with ~400 worms/replicate were analyzed. *p* values were determined from two-way ANOVA, followed by Dunnett's test, or Student's *t*-test. All fold changes were normalized to *EV* control. NS not significant, *$p < 0.05$, **$p < 0.01$, ***$p < 0.001$. In all panels, purple significance marks indicate comparison between *gene(RNAi)* and *EV(RNAi)* in unstressed condition, red marks indicate comparison between *gene(RNAi)* and *EV(RNAi)* in stressed condition (*spg-7(RNAi)*), and black marks indicate comparison between stressed and unstressed conditions.

Unlike the ESRE pathway however, we also observed significant reduction of the basal reporter gene expression following L1 RNAi (**Fig 2A**). Moreover, neither condition completely disrupted UPR^mt activity (compare **Figs 1C and 2A**).

Contrary to what was observed for UPR^mt, *fib-1(RNAi)*, *nol-56(RNAi)*, and *nol-58(RNAi)* significantly increased basal expression level of the *Ptbb-6*::GFP MAPK^mt reporter (**Fig 2C**), indicating that the box C/D snoRNP complex is directly or indirectly involved in repressing its basal expression. This difference disappeared if RNAi targeting the box C/D snoRNP complex was initiated at the L3 stage (**Fig 2D**). RNAi at either stage had no effect on *Ptbb-6*::GFP expression after *spg-7(RNAi)*-mediated induction.

Previously, we demonstrated relationships between the ESRE network and the UPR^mt and MAPK^mt pathways [21]. For example, UPR^mt or MAPK^mt activity normally places a brake on the ESRE network by limiting the production of ESRE-activating ROS. We also identified an ESRE motif in the promoter region of *atfs-1/ATF5* that was required for its full expression. Previously, we used CRISPR to generate a mutant strain where the ESRE site in the promoter region of the *atfs-1* gene was completely removed [21]. This strain was crossed with the UPR^mt reporter *Phsp-6*::GFP reporter and then self-crossed to ensure homozygosity of the ESRE deletion. The resulting strain was used to evaluate the importance of the ESRE site in regulation of *hsp-6*.

We used *spg-7(RNAi)* to induce *Phsp-6*::GFP reporter expression in strains with or without the ESRE motif present in the *atfs-1* promoter reared on vector control or after disrupting the box C/D snoRNP complex. As expected, *spg-7(RNAi)* induced GFP expression in each condition. Similar to our previous findings, basal and induced expression of *Phsp-6*::GFP was reduced in the absence of the ESRE motif from *atfs-1/ATF5* (blue stars in **Fig 3A** (for RNAi starting at L1) **and 3B** (for RNAi starting at L3)). Adding RNAi targeting the box C/D machinery to the ESRE deletion changed basal expression (purple significance marks, compared to basal *EV(RNAi)*) only when RNAi was initiated at L1 stage (**Fig 3A**). Induction of *hsp-6*::*GFP* by *spg-7(RNAi)* was lower when compared to empty vector (red stars) or when compared to induced conditions with an intact promoter and box C/D RNAi (blue stars) (**Fig 3A and 3B**). These data indicate that the box C/D complex regulates UPR^mt both via modulation of the ESRE pathway (due to the ESRE motif, which is required for full expression of *atfs-1*) and independently of ESRE.

Meanwhile, *Ptbb-6*::GFP expression was at least partially dependent upon PMK-3/ MAPK14, as expected, both under wild-type and box C/D snoRNP RNAi conditions (**Fig 3C**). Previously we demonstrated that ATFS-1 plays a role in repressing *tbb-6* expression [21]. Since we showed above that ATFS-1 activity depends on box C/D snoRNPs, we tested whether *fib-1 (RNAi)*, *nol-56(RNAi)*, or *nol-58(RNAi)* would affect ATFS-1-mediated repression of *tbb-6*. In each case, *atfs-1* knockdown was indistinguishable from *atfs-1; snoRNP* double RNAi (**Fig 3D**). Combined, these data argue that the box C/D complex regulates basal expression of the MAPK^mt stress response system, possibly by altering levels of ATFS-1. We also speculate that the absence of changes for basal ESRE reporter expression are due to consistent observations that ESRE network activation must be spurred by recognition of stress, while *Phsp-6*::GFP and *Ptbb-6*::GFP exhibit low levels of expression even in the absence of stress [18].

To ensure that the simultaneous use of two RNAi targets did not reduce their independent knockdown efficiency, several supplemental experiments were performed. First, empty vector controls were substituted with an RNAi construct targeting luciferase. As none of the strains in use encode luciferase, expression of this RNAi construct will engage the RNAi machinery, but should not have any other biological effect. As expected, no observable difference was seen in expression of the *3xESRE*::GFP, *Phsp-6*::GFP, or *Ptbb-6*::GFP reporters when the luciferase construct was substituted for the empty vector, either alone or in combination with RNAi

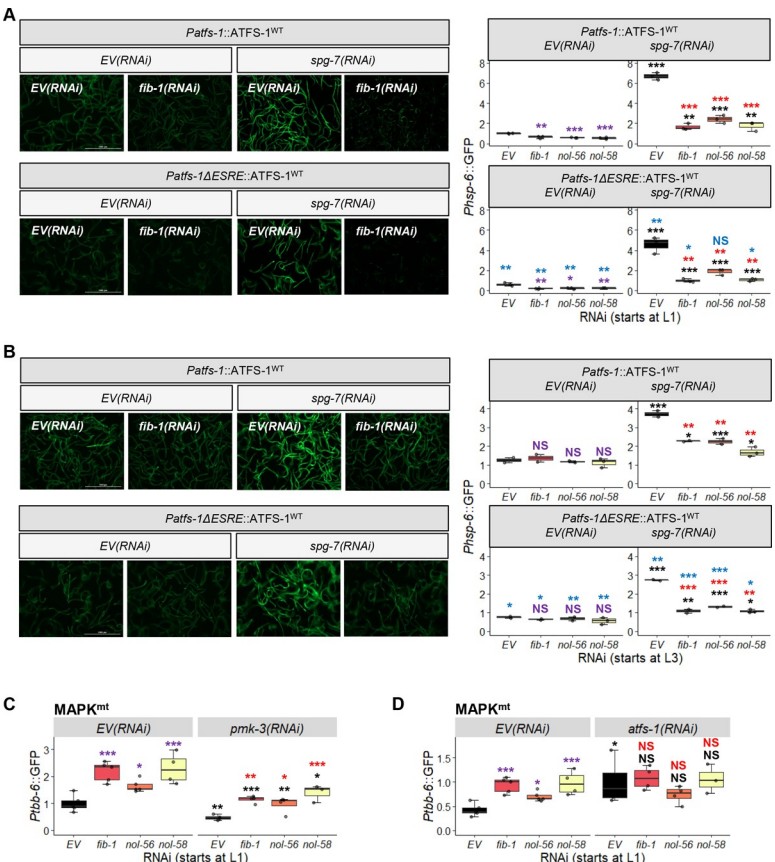

**Fig 3. Disruption of box C/D snoRNPs affects UPR$^{mt}$ and MAPK$^{mt}$ expression through the ESRE motif. (A-B)** Images and **(A-D)** quantification of GFP fluorescence of *C. elegans* carrying *Phsp-6*::GFP **(A, B)** or *Ptbb-6*::GFP **(C, D)** reporters that were reared on *E. coli* expressing empty vector (*EV*) or RNAi targeting *fib-1/FBL*, *nol-56/Nop56*, or *nol-58/Nop58*. **(A, B)** *Phsp-6*::GFP reporter strains were wild-type or crossed with *Patfs-1ΔESRE*::ATFS-1$^{WT}$. RNAi was paired with empty vector (*EV*) or *spg-7(RNAi)* **(A, B)**, *pmk-3(RNAi)* **(C)**, or *atfs-1(RNAi)* **(D)**. Three biological replicates with ~400 worms/replicate were analyzed. **(A, B)** Representative images are shown on the left. *p*-values were determined from two-way ANOVA, followed by Dunnett's test, or Student's *t*-test. All fold changes were normalized to *EV* control. NS not significant, $^*p < 0.05$, $^{**} p < 0.01$, $^{***} p < 0.001$. In all panels, purple significance marks indicate comparison between *gene(RNAi)* and *EV(RNAi)* in unstressed condition, red marks indicate comparison between *gene (RNAi)* and *EV(RNAi)* in stressed condition (*spg-7(RNAi)*), and black marks indicate comparison between stressed and unstressed conditions. Additionally, blue significance marks indicate comparison between *Phsp-6*::GFP expression in *Patfs-1ΔESRE*::ATFS-1$^{WT}$ and wild-type backgrounds.

targeting other genes (**S2A and S2B Fig**). Additionally, *fib-1* and *spg-7* transcripts were quantified via qRT-PCR when they were knocked down either alone or in combination with luciferase RNAi (**S2C and S2D Fig**).

We further asked whether localization of these snoRNPs in the nucleolus was necessary for regulation. We knocked down *ruvb-1/RUVB*, an AAA+ ATPase that promotes box C/D snoRNP assembly and localization to nucleoli [47]. Although worms reared on *ruvb-1(RNAi)-* expressing *E. coli* at L1 did not show growth arrest, *ruvb-1/RUVB(RNAi)* markedly reduced ESRE expression following stress (**S3A Fig**). However, induced expression of UPR$^{mt}$ was not affected (**S3B Fig**) and increased in basal expression of MAPK$^{mt}$ reporter was also more modest than what was observed for C/D snoRNP knockdown (**S3C Fig**). This suggests that localization of box C/D snoRNPs may affect some but not all the responses of mitochondrial surveillance.

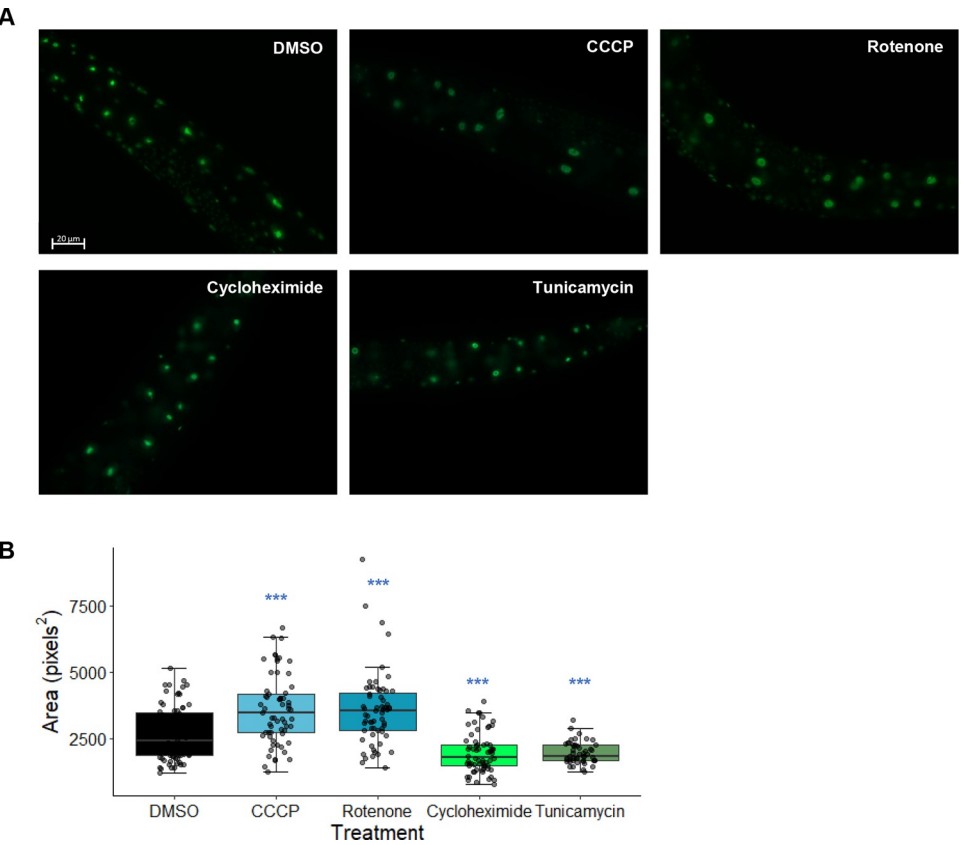

**Fig 4. Mitochondrial disruption increases the size of *Pfib-1*::FIB-1::eGFP fluorescent punctae. (A)** Images and **(B)** quantification of FIB-1::eGFP in worms treated for 4 hours with vehicle (DMSO), 10 μM CCCP, 50 μM rotenone, 2 mg/mL cycloheximide, or 60 μM tunicamycin. Three biological replicates with ~25 worms/replicate were analyzed. *p*-values were determined from one-way ANOVA, followed by Dunnett's test. *** $p < 0.001$.

*ruvb-1/RUVB(RNAi)* in a worm strain carrying an endogenously expressed, GFP-tagged FIB-1 (*Pfib-1*::FIB-1::eGFP) was tested to determine the expression and localization of FIB-1/FBL. Knockdown of *fib-1/FBL* starting at L1 or L3 significantly reduced FIB-1 fluorescence intensity (**S3D Fig**). Similarly, *ruvb-1/RUVB(RNAi)* significantly reduced FIB-1/FBL fluorescence, but to a lesser extent than *fib-1* knockdown (**S3D Fig**). This is consistent with our observation that ESRE expression was reduced more in *fib-1(RNAi)* compared to *ruvb-1(RNAi)*. Interestingly, we also observed that *Pfib-1*::FIB-1::eGFP punctae increased in size after treatment with the mitochondrial poisons CCCP or rotenone (**Fig 4A and 4B**). In contrast, treatment with cycloheximide (a translational inhibitor) and tunicamycin (an ER stress inducer) reduced the size of *Pfib-1*::FIB-1::eGFP punctae (**Fig 4A and 4B**). This shows that FIB-1/FBL fluorescence is altered under ESRE-activating conditions and may indicate that mitochondrial damage changes either the expression level of the gene or alters the subcellular localization of the protein product.

## Disruption of box H/ACA snoRNP machinery does not affect mitochondrial surveillance pathways

One possible explanation for the phenomena that we observed was that the reduction of 2'-O-methylation of rRNA compromised normal ribosomal function, and that this reduced ESRE gene expression. If this were the case, other broad-scale ribosomal changes should have similar

outcomes. For example, disrupting box H/ACA-mediated conversion of dozens to hundreds of uridine residues to pseudouridine in rRNA [50,51] should have a very similar effect. In contrast to box C/D snoRNP genes, RNAi targeting *nola-3/Nop10*, *Y48A6B.3/Nhp2*, or *Y66H1A.4/Gar1* showed no significant change for any of the mitochondrial surveillance pathways tested, whether under basal or mitochondrial stress-induced conditions (**S4 Fig**). This result indicates that the function of box C/D snoRNPs in regulating mitochondrial homeostasis is specific. It also suggests that wide-scale disruption of the ribosome is unlikely to be the cause of diminished ESRE activation. mRNA levels were quantified using qRT-PCR to ensure that RNAi treatment had the intended effect (**S5 Fig**).

## Suppressing translation does not recapitulate changes in mitochondrial surveillance caused by disrupting the box C/D snoRNP complex

2'-*O*-methylation has a number of effects on ribosome maturation and stability. One possible consequence of disrupting ribosomal biology is a global reduction in translation [52,53]. Translation efficiency is known to be a target of surveillance in *C. elegans*, as shown by increased expression of the immune response gene *irg-1* after exposure to exotoxin A, hygromycin, or cycloheximide [1,2]. Recently it was shown that *fib-1/FBL* knockdown also activates *irg-1* [42]. We recapitulated these data and observed increased *Pirg-1*::GFP reporter expression on *fib-1(RNAi)*, *nol-56(RNAi)* or *nol-58(RNAi)* as well as after treatment with cycloheximide (**S6 Fig**). Consequently, Tiku *et al*, hypothesized that innate immune activation was an indirect event, caused by a diminished translation as a result of fibrillarin disruption and compromised ribosomal function [42]. We set out to explore whether the decreased levels of the ESRE reporter following *fib-1(RNAi)* are caused by the same mechanism.

We used RNAi to target the genes encoding components of the eukaryotic 48S transcription initiation complex, including *clu-1/eIF3A*, *inf-1/eIF4A*, *ife-2/eIF4E*, *ifg-1/eIF4G*, and *T12D8.2/eIF4H*. Since RNAi targeting *ifg-1/eIF4G* and *inf-1/eIF4A* compromised development when RNAi was started at the L1 stage, subsequent experiments were performed by feeding RNAi starting at the L3 stage of development. Expression of reporters for the mitochondrial surveillance pathways being tested was not consistently affected by disrupting the 48S complex (**Fig 5**), with the exception of *inf-1/eIF4A(RNAi)*, which reduced rotenone-mediated ESRE activation and *spg-7(RNAi)*-mediated UPR$^{mt}$ activation, and increased basal expression of MAPK$^{mt}$. However, induction of the *Phsp-6*::GFP reporter by *spg-7(RNAi)* was also significantly decreased for *clu-1/eIF3A(RNAi)*, *ifg-1/eIF4G(RNAi)*, *and ife-2/eIF4E(RNAi)*, suggesting some specialized interactions (**Fig 5B**).

As a final test, we treated reporter worms for each of the three mitochondrial surveillance pathways with the chemical translational inhibitor cycloheximide. Under conditions where *irg-1* activation was observed (indicating successful disruption of translation, see **S6B Fig**), cycloheximide exposure had no effect on ESRE or MAPK$^{mt}$ reporter expression (**S7 Fig**). These results further indicated that general translational reduction is unlikely to be the mechanism underlying box C/D snoRNP regulation of the ESRE mitochondrial surveillance network.

## Box C/D snoRNPs repress innate immune responses

As mentioned above, fibrillarin knockdown has previously been linked to increased pathogen resistance [42], and we and others observed increased expression of *Pirg-1*::GFP, an immune reporter, in *fib-1(RNAi)* worms. To test whether disruption of the box C/D snoRNP complex induced other cellular defense pathways, we monitored the expression of *Pirg-5*::GFP reporter, which is activated by a number of pathogens and xenobiotics [54,55] and is relatively insensitive to translational inhibition (**Fig 6A** and [2]).

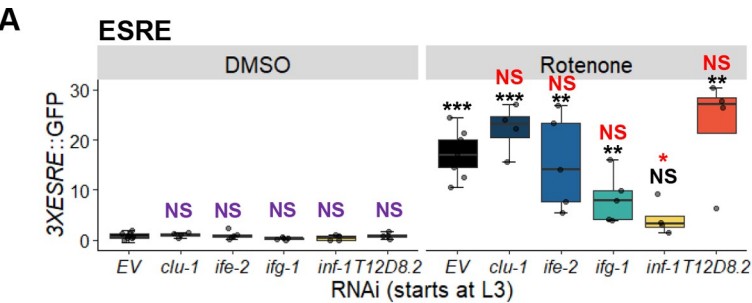

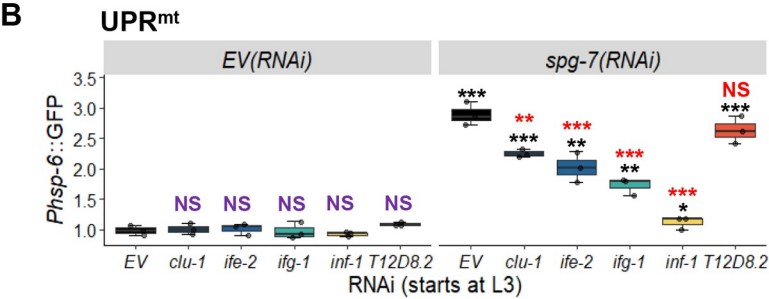

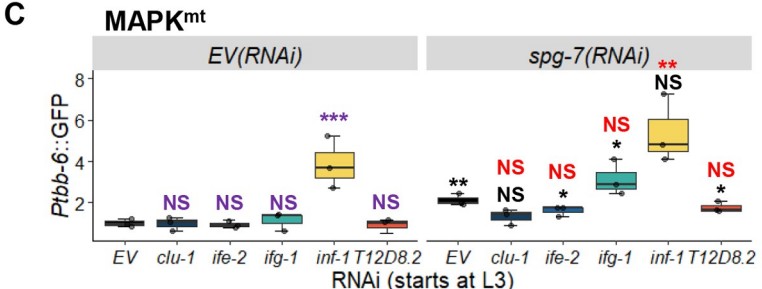

**Fig 5. Blocking translational initiation generally does not diminish activation of mitochondrial surveillance pathways.** Quantification of GFP fluorescence of *C. elegans* carrying *3XESRE*::GFP **(A)**, *Phsp-6*::GFP **(B)**, or *Ptbb-6*::GFP **(C)** reporters that were reared on *E. coli* expressing empty vector (*EV*) or RNAi targeting the eukaryotic initiation factors *clu-1/eIF3A*, *ife-2/eIF4E*, *ifg-1/eIF4G*, *inf-1/eIF4A*, or *T12D8.2/eIF4H*. **(A)** Worms were treated for 8 hours with vehicle (DMSO) or 50 μM rotenone. **(B, C)** Worms were stressed with *spg-7(RNAi)*, and an empty vector (*EV*) control was included. Three biological replicates with ~400 worms/replicate were analyzed. *p*-values were determined from two-way ANOVA, followed by Dunnett's test, or Student's *t*-test. All fold changes were normalized to DMSO-*EV* or *EV* control. NS not significant, * $p < 0.05$, ** $p < 0.01$, *** $p < 0.001$. In all panels, purple significance marks indicate comparison between *gene(RNAi)* and *EV(RNAi)* in unstressed condition (DMSO or *EV(RNAi)*), red marks indicate comparison between *gene(RNAi)* and *EV(RNAi)* in stressed condition (rotenone or *spg-7(RNAi)*), and black marks indicate comparison between stressed and unstressed conditions.

Worms carrying the *Pirg-5*::GFP reporter were reared on either empty vector or RNAi targeting components of the box C/D snoRNP complex, and then GFP expression was evaluated in young adults. Interestingly, this disruption activated *irg-5* more strongly than *P. aeruginosa* infection on agar in empty vector controls (**Fig 6B**). It is also worth noting that infecting worms with *fib-1(RNAi)* or *nol-58(RNAi)* with *P. aeruginosa* did not further increase GFP expression compared to uninfected counterparts, suggesting that *irg-5* induction may have already been maximized.

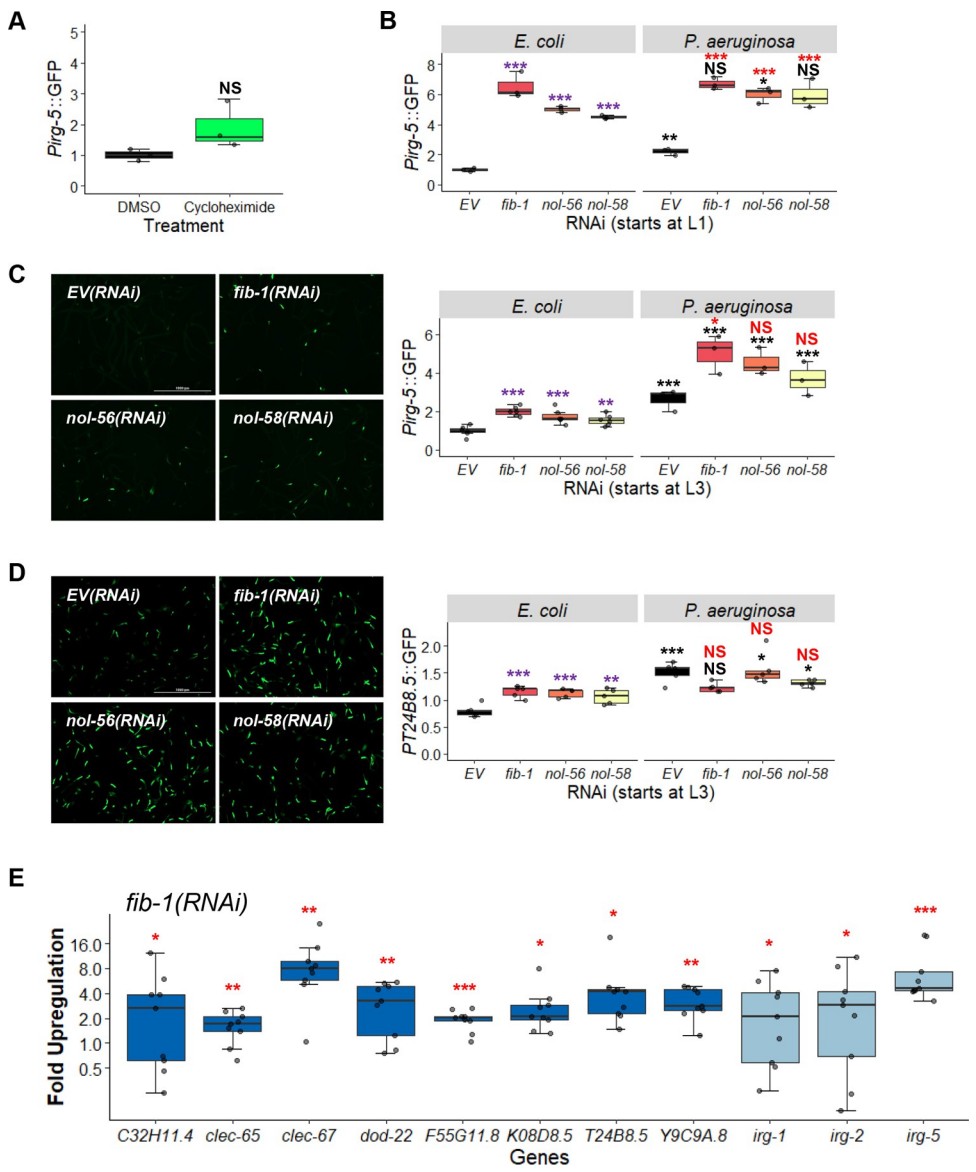

**Fig 6. Knockdown of box C/D snoRNPs increases immune activation.** Quantification of GFP fluorescence of *C. elegans* carrying *Pirg-5*::GFP (**A-C**) or *PT24B8.5*::GFP (**D**) reporter. (**A**) Worms were treated for 8 hours with vehicle (DMSO) or translation elongation inhibitor cycloheximide (2 mg/mL). (**B-D**) Worms were reared on *E. coli* expressing empty vector (*EV*), *fib-1(RNAi)*, *nol-56(RNAi)*, or *nol-58(RNAi)*. Young adult worms were transferred onto plates containing *E. coli* or *P. aeruginosa* for 8 hours before imaging. RNAi treatment was started at the L1 (**B**) or L3 (**C, D**) stage. Representative images are shown on the left for unstressed conditions (**C, D**). (**E**) Fold-upregulation of immune genes under the regulation of both UPR$^{mt}$ and p38 MAPK pathways (dark blue) or genes under the control of other regulatory pathways (light blue) in wild-type worms reared on *fib-1(RNAi)* from the L3 stage. Three biological replicates with ~400 (**A-D**) or ~8,000 (**E**) worms/replicate. *p*-values were determined from two-way ANOVA, followed by Dunnett's test, or Student's *t*-test for GFP quantification assays, or one-way ANOVA, followed by Dunnett's test for qRT-PCR. All fold-changes were normalized to *EV* control on *E. coli*. NS not significant, *$p < 0.05$, ** $p < 0.01$, *** $p < 0.001$. In panels with multi-colored significance marks, purple indicates comparison between *gene(RNAi)* and *EV (RNAi)* in the control condition (*E. coli*), red indicates comparison between *gene(RNAi)* and *EV(RNAi)* after exposure to *P. aeruginosa*, and black indicates comparison between stressed and control conditions.

As we had previously observed, box C/D RNAi might affect pathway induction differently when RNAi was started at L1 vs L3. Specifically, increased *irg-5* basal induction was much stronger when RNAi feeding was initiated at the L1 stage (**Fig 6B and 6C**). This observation is also

consistent with our interpretation that *Pirg-5*::GFP expression is nearly saturated when box C/D is knocked down early in development (**Fig 6B**). These results indicate that early developmental C/D snoRNP complexes are required for appropriate innate immune function later in development.

Recently, Campos et al. established a link between mild mitochondrial impairment and enhanced immunity [32]. In their report, regulation of several immune genes was dependent on both the UPR$^{mt}$ and the PMK-1/p38 immune pathways. To address this possibility, we tested whether the loss of box C/D snoRNPs affected expression of p38 MAPK immune effectors. First, expression of the *PT24B8.5*::GFP reporter strain was tested after box C/D RNAi by exposing the worms to *E. coli* or *P. aeruginosa*. *T24B8.5* is an innate immune effector gene under the control of both PMK-1 and the UPR$^{mt}$. Under these conditions, we saw that box C/D RNAi activated innate immune pathways on normal laboratory food for the host, but did not drive it to higher levels than exposure to a pathogen (**Fig 6D**). A wider range of immune effectors were also assessed via qRT-PCR after *fib-1(RNAi)* and were shown to also be significantly upregulated (**Fig 6E**). This result strongly argues that the loss of box C/D snoRNPs causes a broad-spectrum increase in innate immune activation.

As *irg-5* was not significantly upregulated by translational repression, snoRNPs are likely to affect innate immune pathways via multiple mechanisms. *irg-5* is known to be controlled by several transcriptional regulators, including PMK-1/p38 MAPK and ATF-7/ATF7 [55], both of which are established regulators of innate immunity in *C. elegans* [56]. ATF-7/ATF7 functions downstream of PMK-1/p38 MAPK, but it can regulate *irg-5* activation independently of PMK-1/p38 MAPK. NHR-86/HNF4 also regulates *irg-5* expression, specifically in response to the small molecule immune stimulant RPW-24 [57]. To determine whether any of these transcriptional regulators were involved in box C/D regulation of innate immunity, we compared *Pirg-5*::GFP expression in worms with double RNAi targeting *fib-1/Fibrillarin*, *nol-56/Nop56*, or *nol-58/Nop58* and *pmk-1/p38 MAPK*, *atf-7/ATF7*, or *nhr-86/HNF4*.

Both *pmk-1(RNAi)* (**Fig 7A**) and *atf-7(RNAi)* (**Fig 7B**) reduced *Pirg-5*::GFP expression, with the latter virtually abolishing reporter fluorescence. *nhr-86(RNAi)* (**Fig 7C**) had no apparent effect. These data suggest that the effects of the box C/D snoRNP complex on *irg-5* are upstream of its known regulation by the transcription factor ATF-7/ATF7. However, *atf-7(RNAi)* did not affect expression of *irg-1*, an immune effector whose basal expression was also upregulated upon box C/D snoRNP knockdown (**Fig 7D**). This suggests that box C/D snoRNPs exhibit complex modulation of innate immune pathways. We verified this result using qRT-PCR to measure several effectors dependent upon PMK-1 and ATF-7 in *pmk-1 (km25)* and *atf-7(gk715)* mutants reared on *fib-1(RNAi)*. Generally, *fib-1(RNAi)*-induced expression of these immune genes were dramatically reduced in both mutants, although *irg-1* was an exception (**Fig 7E**). Notably, the *pmk-1(km25)* mutant also completely abolished expression of *irg-5*, likely due to a complete loss of PMK-1 in the mutant.

This led us to question whether ATF-7 is involved in the regulation of other reporters that are differentially expressed under box C/D disruption. To test the *3XESRE*::GFP reporter, worms were reared on combinations of empty vector, *fib-1(RNAi)*, *atf-7(RNAi)*, or both starting at the L3 stage. Young adults were then exposed to rotenone and reporter induction was measured. Knockdown of *atf-7* was indistinguishable from vector control (red "NS" mark). *atf-7(RNAi); fib-1(RNAi)* showed no apparent difference from just *fib-1(RNAi)* (blue "NS" mark) (**Fig 7F**). This indicates that the box C/D regulation of ESRE is independent of ATF-7 and is different from the regulation of *irg-5*.

We also tested whether the increase in basal expression of *Ptbb-6*::GFP after *fib-1(RNAi)* was related to ATF-7 by rearing the reporter on *fib-1(RNAi)* alone or with *atf-7(RNAi)* starting at the L1 larval stage. As expected, basal levels of the reporter were increased after *fib-1(RNAi)* but not by *atf-7(RNAi)* alone (**Fig 7G**). However, we did observe an additive effect on *Ptbb-6*::

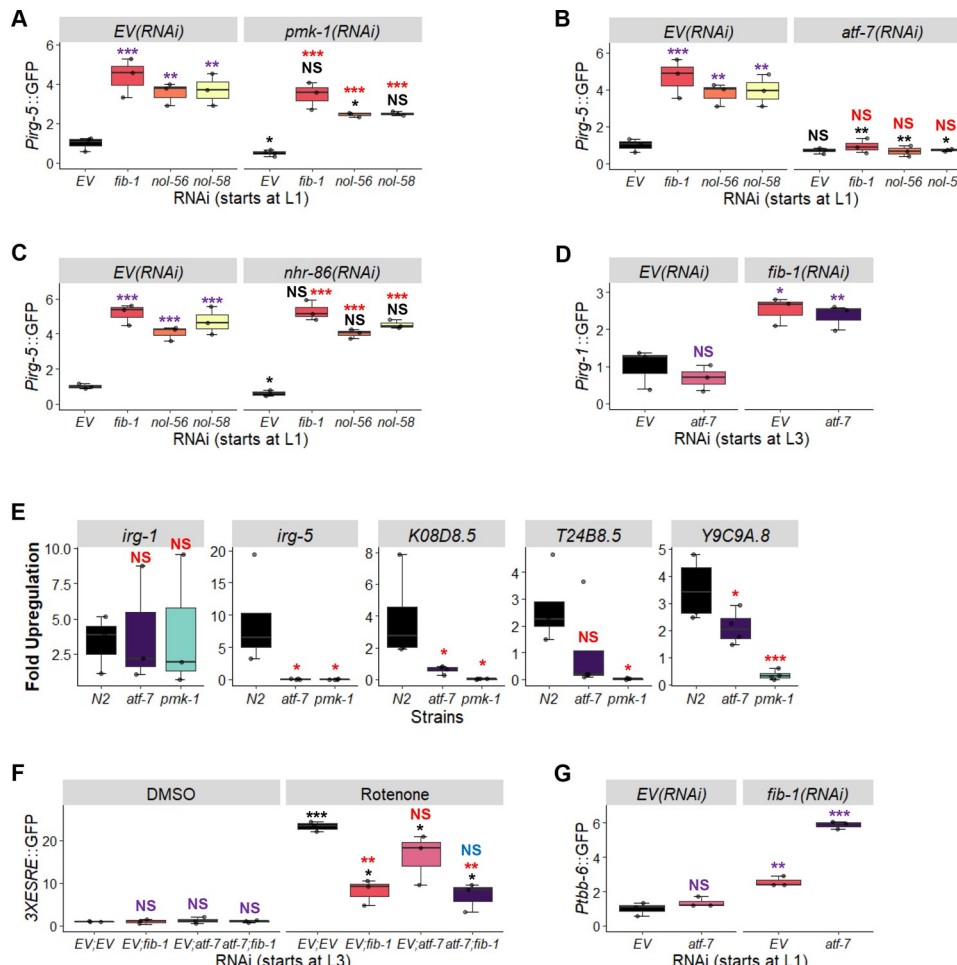

**Fig 7. ATF-7 and PMK-1 partially mediate immune responses induced by the loss of C/D snoRNPs but do not affect mitochondrial surveillance pathways. (A-C)** Innate immune activation was quantified by measuring GFP expression in *Pirg-1::GFP* worms reared on empty vector (*EV*), *fib-1(RNAi)*, *nol-56(RNAi)*, or *nol-58(RNAi)* in all pair-wise combinations with empty vector (*EV*) (A-C), *pmk-1(RNAi)* (A), *atf-7(RNAi)* (B), or *nhr-86(RNAi)* (C). **(D)** Innate immune activation was quantified by measuring GFP expression in *Pirg-1::GFP* worms reared on the following RNAi combinations: *EV; EV, atf-7; EV, EV; fib-1*, or *atf-7; fib-1*. **(E)** Expression of the innate immune genes *irg-1*, *irg-5*, *K08D8.5*, *T24B8.5*, and *Y9C9A.8* was measured via qRT-PCR in wild-type (N2), *atf-7(gk715)*, or *pmk-1(km25)* mutants reared on empty vector or *fib-1(RNAi)*. Gene expression was compared to empty vector controls. **(F,G)** GFP fluorescence was quantified in worms carrying *3XESRE::GFP* (F) or *Ptbb-6::GFP* (G) reporters reared on *EV; EV, atf-7 (RNAi); EV, EV; fib-1(RNAi)*, or *atf-7(RNAi); fib-1(RNAi)*. In (F), worms were exposed to either DMSO (control) or 50 μM rotenone for 8 hours prior to imaging. RNAi treatment was started at the L1 stage (A-C, G) or L3 stage (D-F). Three biological replicates with ~400 (GFP quantification) or ~8,000 (qRT-PCR) worms/replicate were used. *p*-values were determined from two-way ANOVA, followed by Dunnett's test, or Student's *t*-test for GFP quantification assay, or one-way ANOVA, followed by Dunnett's test for qRT-PCR data. NS not significant, $^*p < 0.05$, $^{**} p < 0.01$, $^{***} p < 0.001$. In panels with multi-colored significance marks, purple indicates comparison between *gene(RNAi)* and *EV (RNAi)* in control (*EV(RNAi)* (A-C) or DMSO (F)) conditions. Red indicates comparison between *gene(RNAi)* and *EV (RNAi)* in *pmk-1(RNAi)* (A), *atf-7(RNAi)* (B), *nhr-86(RNAi)* (C), or rotenone (F). Black indicates comparison between *pmk-1(RNAi)* (A), *atf-7*(RNAi) (B), *nhr-86*(RNAi) (C), or rotenone (F) to untreated controls in the same panel. Blue indicates comparison between *atf-7(RNAi);fib-1(RNAi)* and *EV;fib-1(RNAi)* after rotenone treatment (F).

*GFP* in *atf-7(RNAi);fib-1(RNAi)*, suggesting that the two genes work together to limit inappropriate expression of the MAPK$^{mt}$ pathway.

Although our previous experiments had ruled out a role for box H/ACA snoRNPs in the regulation of mitochondrial surveillance (S4 Fig), we wanted to investigate their role in innate

immunity. A knockdown of box H/ACA did not invoke any response from the immune genes *irg-1* and *irg-5* (**S8 Fig**), confirming the specificity of box C/D snoRNPs.

## Box C/D snoRNPs provide protection against mitochondrial damage from pathogens

To test whether the loss of box C/D snoRNPs had physiologically relevant consequences, we performed *P. aeruginosa* Liquid Killing (liquid pathogenesis) and Slow Killing (agar pathogenesis) assays to measure survival. Although both assays use the same pathogen, the virulence and pathogenic mechanisms and the host defenses differ. In the Liquid Killing assay, host death occurs due to the production of the siderophore pyoverdine, which is secreted by the bacterium to obtain iron [26,58]. Pyoverdine enters host tissue and removes iron from mitochondria, causing sufficient damage to inflict death [59,60]. This damage also activates the ESRE mitochondrial surveillance network, which is important for host defense [20].

The Slow Killing assay involves a more traditional form of bacterial pathogenesis, where the host intestine is colonized by the pathogen. Killing involves quorum sensing, although the precise cause of death has not yet been determined [61,62]. Slow Killing activates the conventional NSY-1/SEK-1/PMK-1 MAPK pathway, which is the most common antibacterial defense in *C. elegans* [63–65]. Interestingly, there appears to be little overlap between the two defense networks, as there is no activation of ESRE during Slow Killing and PMK-1 activity is detrimental for survival in Liquid Killing conditions [25,66].

Worms were reared on RNAi targeting *fib-1/Fibrillarin*, *nol-56/Nop56*, or *nol-58/Nop58* from the L3 larval stage, and then young adults were exposed to *P. aeruginosa* strain PA14 either under Liquid Killing or Slow Killing conditions. As anticipated based on the role of ESRE in improving survival in Liquid Killing and the observation that box C/D snoRNP knockdown compromises ESRE function, removal of box C/D function strongly reduced host survival in Liquid Killing (**Fig 8A**). Interestingly, we saw the opposite in Slow Killing, where box C/D snoRNP RNAi slightly, but statistically significantly, increased host survival (**Fig 8B and S3 Table**). This is consistent with a prior report that FIB-1 reduces host survival during infection [42]. We also quantified intestinal colonization in this model and observed significantly lower colonization of the worms' intestines when box C/D snoRNP complexes were disrupted (**Fig 8C**).

Next, the effect of ATF-7 and PMK-1 on increased survival in Slow Killing by box C/D RNAi was assessed. Slow Killing assays were performed with wild-type, *atf-7(gk715)*, and *pmk-1(km25)* mutant worms reared on empty vector or box C/D RNAi. Interestingly, the loss of box C/D snoRNPs increased survival of *atf-7(gk715)* and *pmk-1(km25)* mutants, as was observed for wild-type worms (**Figs 8D and S9 and S4 Table**). These results were not entirely unexpected, considering that there are multiple immune pathways in the host and some innate immune genes are outside of the control of the MAPK pathway [67,68]. Together, these results strengthen our hypothesis that the loss of box C/D snoRNPs increased host immune responses.

Interestingly, Liquid Killing also increased the size of FIB-1::eGFP punctae (**Fig 8E**), reinforcing the conclusion above that the change in FIB-1::GFP patterns may result from mitochondrial damage. In contrast, infection with *P. aeruginosa* in the agar pathogenesis model (Slow Killing) reduced FIB-1::GFP punctae size (**Fig 8F**). This is consistent with a previous report that bacterial infection reduced both FIB-1 level and nucleolar size [42].

## Box C/D snoRNP support normal mitochondrial function

To assess the importance of the box C/D snoRNPs in mitochondrial function more directly, we measured mitochondrial membrane potential, mitochondrial mass, and oxygen

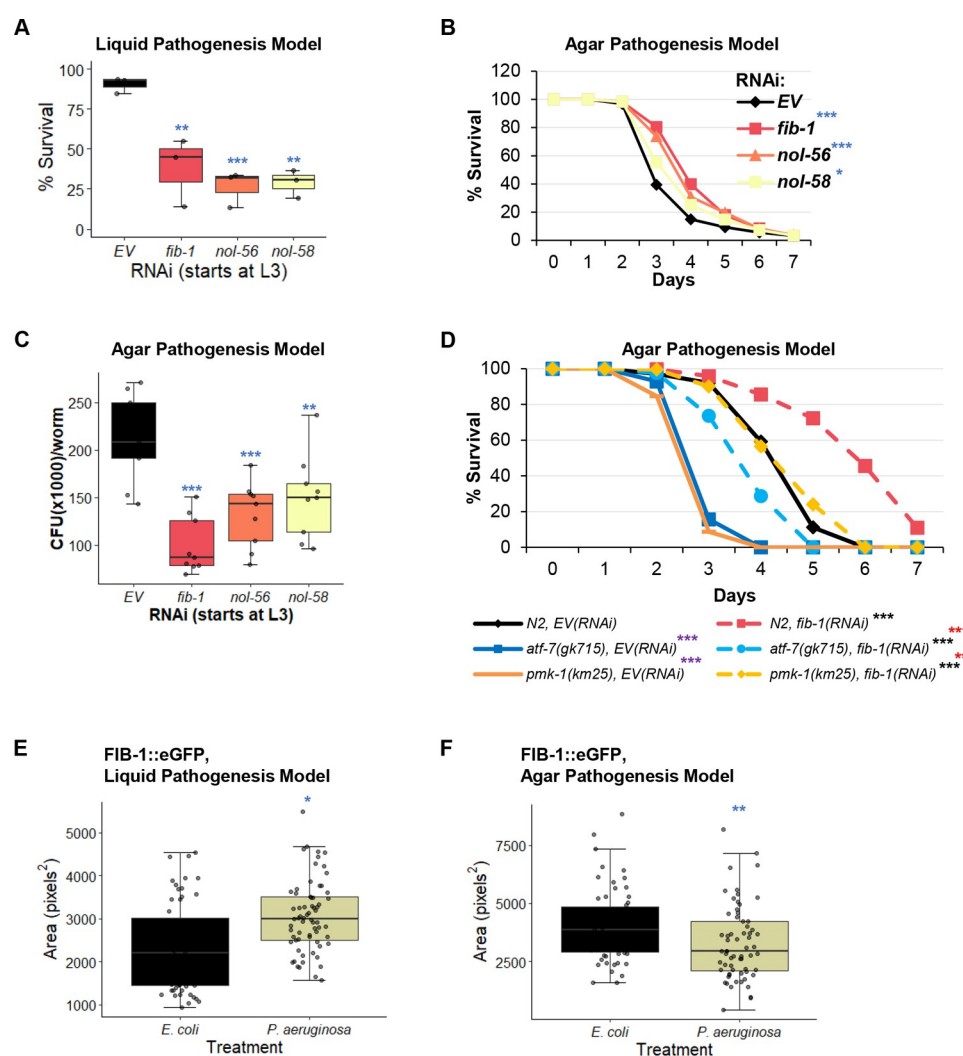

**Fig 8. Box C/D snoRNP disruption has context-specific effects on *P. aeruginosa* pathogenesis. (A, B)** Survival of *glp-4(bn2)* worms grown on *box C/D(RNAi)* **(A)** Liquid Killing (LK, liquid pathogenesis model) and **(B)** Slow killing (SK, agar pathogenesis model) assays. **(C)** Quantification of colony forming units (CFU) within the intestine of *glp-4 (bn2)* worms grown on *box C/D(RNAi)* 48 hours post infection with *P. aeruginosa* in the SK assay. **(D)** Survival of N2 wild-type, *atf-7(gk715)*, or *pmk-1(km25)* mutant worms reared on empty vector (*EV*) or *fib-1(RNAi)*. **(E, F)** Quantification of fluorescence of worms carrying a *Pfib-1*::FIB-1:eGFP reporter 10 hours after exposure to *P. aeruginosa* in Liquid Killing (LK, **(E)**) or Slow Killing (SK, **(F)**) pathogenesis models. Three biological replicates with ~400 (LK), ~150 worms/replicate (SK), or ~60 (CFU assay and FIB-1::eGFP imaging) worms/replicate were analyzed. Representative replicates are shown for the SK assays. *p*-values were determined from one-way ANOVA, followed by Dunnett's test for LK or CFU assay, log-rank test for SK, or Student's *t*-test for FIB-1::eGFP imaging data. $^*p < 0.05$, $^{**}p < 0.01$, $^{***}p < 0.001$. **(D)** Purple significance marks indicate comparison between *atf-7(gk715)* or *pmk-1(km25)* mutants and wild-type (N2) worms reared on empty vector RNAi (*EV*), red marks indicate comparison between *atf-7 (gk715)* and *pmk-1(km25)* mutants to wild-type (N2) worms reared on *fib-1(RNAi)*, and black marks indicate comparison between *fib-1(RNAi)* and *EV(RNAi)* for each worm strain.

consumption rate in wild-type worms reared on *E. coli* carrying either an empty vector control or RNAi targeting box C/D snoRNPs. Mitochondrial membrane potential was measured using MitoTracker Red, a dye that accumulates in mitochondria in a membrane-potential-dependent fashion. Mitochondrial mass was measured using MitoTracker Green, a dye that is comparatively insensitive to membrane potential. Oxygen consumption rate was measured using a biological oxygen monitor with a Clark-type oxygen electrode. Consistent with our hypothesis,

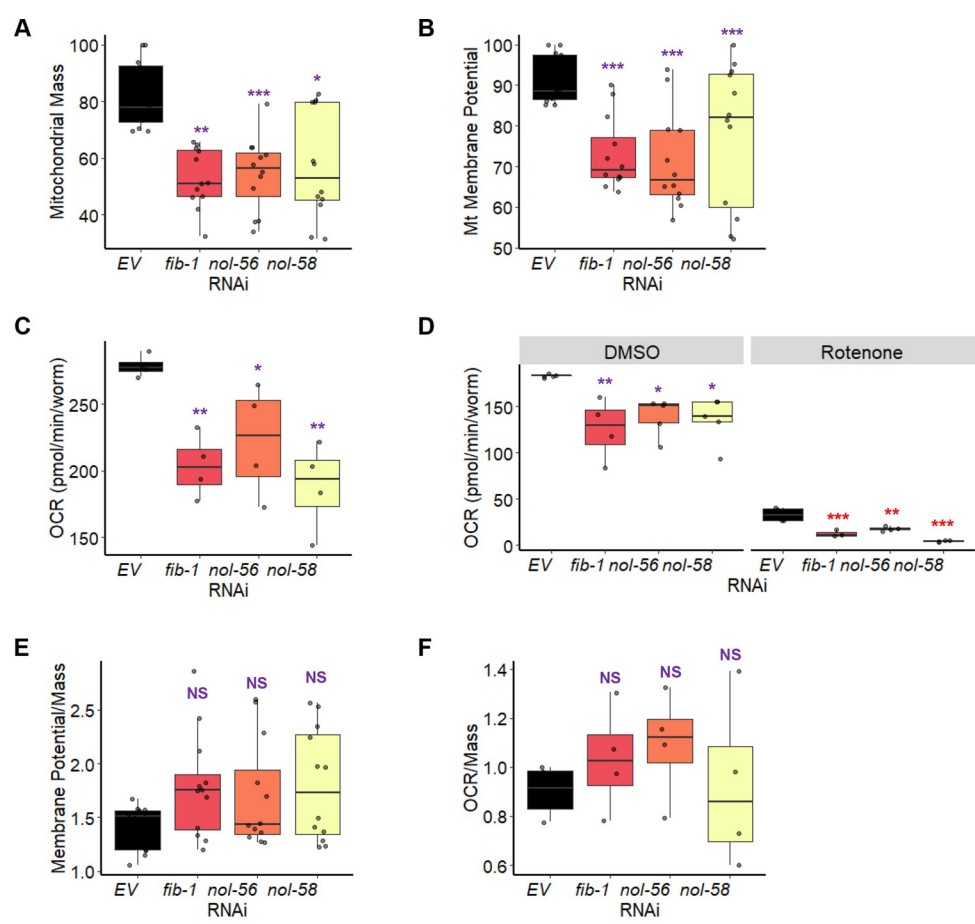

**Fig 9. The loss of box C/D snoRNPs reduced mitochondrial health and function.** Measurement of **(A)** mitochondrial mass, **(B)** mitochondrial membrane potential, and **(C-D)** oxygen consumption rate of wild-type (N2) worms reared on *E. coli* expressing RNAi targeting empty vector (*EV*) or the box C/D snoRNP machinery (*fib-1/FBL*, *nol-56/Nop56*, or *nol-58/Nop58*). Mitochondrial parameters were measured on basal levels **(A-C)** or after 8 hours treatment with 25 μM rotenone or DMSO control **(D)**. **(E)** Mitochondrial membrane potential or **(F)** oxygen consumption rate was normalized to mitochondrial mass. Three biological replicates with ~400 (mitochondrial mass and membrane potential measurements) or 6,000 (oxygen consumption rate measurements) worms/replicate were analyzed. *p* values were determined from one-way or two-way ANOVA, followed by Dunnett's test. * $p < 0.05$, ** $p < 0.01$, *** $p < 0.001$. **(D)** Purple significance marks indicate comparison between *fib-1(RNAi)*, *nol-56(RNAi)*, or *nol-58(RNAi)* and *EV(RNAi)* in DMSO, red significance marks indicate comparison between *fib-1(RNAi)*, *nol-56 (RNAi)*, or *nol-58(RNAi)* and *EV(RNAi)* in rotenone.

we observed reduction in all three parameters after box C/D snoRNP disruption (**Fig 9A–D**). It is worth noting that the decrease in mitochondrial membrane potential and oxygen consumption rate was proportional to the decrease in mitochondrial mass (**Fig 9E and 9F**). These data further strengthen our conclusion that disruption of box C/D snoRNPs dysregulates mitochondrial surveillance pathways and compromises normal mitochondrial biology.

## Discussion

In this study, we identified a role for the box C/D snoRNP complex in regulating the switch between mitochondrial surveillance and innate immunity. Using biochemical approaches, we made the unexpected discovery that FIB-1/Fibrillarin and NOL-56/Nop56 were associated with a tandem repeat of the ESRE motif. This is unexpected for two reasons. First, the RNA sequences for which box C/D snoRNAs are named, the C (**RUGAUGA**) and D (**CUGA**)

boxes, have been well-studied and bear very little sequence similarity to the ESRE motif (**TCTGCGTCTCT**). Although these are all consensus sequences and have some variability, they have essentially no match, making it seem quite unlikely that the proteins are recognizing the ESRE site directly. Second, as noted above, the association of NOL-56/Nop56 with the box C/D snoRNP complex is based on protein-protein interactions, suggesting that the interaction with the ESRE bait is likely to involve the entire snoRNA complex.

Interestingly, the interactions between box C/D snoRNPs and mitochondrial surveillance pathways are specific. RNAi of box C/D starting at L3 stage results in decreased expression of reporters for ESRE and UPR^mt, while expression of MAPK^mt is not affected. Importantly, knockdown of gene expression at this stage does not decrease worm size or shorten lifespan, suggesting that the health of the worms is not seriously compromised. This indicates that the reduction in ESRE or UPR^mt signaling is not an artifact of reduced reporter expression due to worm malaise. Instead, the interaction is likely to be specific. This is further validated by our observations that reducing ESRE expression via box C/D RNAi in the context of Liquid Killing shortens lifespan and that box C/D RNAi disrupts mitochondrial function.

There are some caveats to our use of RNAi in this study of the role of box C/D snoRNPs. First, RNAi exhibits different efficacy in knockdown from gene-to-gene, making it possible that different members of the complex may be differently affected [69]. RNAi also occasionally yields different phenotypes than mutants [70]. For example, box C/D snoRNP RNAi induces *irg-5* immune gene expression. Both *pmk-1(RNAi)* and the *pmk-1(km25)* mutant have reduced *irg-5* expression in this case, but the mutants had a greater impact. Despite these caveats, our findings support the conclusions that we have drawn.

Box C/D snoRNPs have recently been linked to an increasing variety of function beyond their canonical role in rRNA methylation, including rare cases of guiding RNA editing [71], tRNA methylation [72], and even association with mRNA (including an 'orphan' snoRNA with no known rRNA target that destabilizes several mRNAs) [73,74]. Given recent observations that snoRNAs are more frequently found in the cytoplasm after exposure to oxidative stress or heat shock [75–77], one possible explanation is that snoRNAs could be leaving the nucleolus to regulate ESRE genes by methylating mRNAs.

Although this idea is intriguing, it appears to be mathematically implausible as a broad explanation for the mechanism of box C/D regulation of ESRE gene expression. Only approximately 50 box C/D snoRNAs are predicted in the genome of *C. elegans*. In contrast, the ESRE nucleotide motif is present in the promoter region of ~8% of all predicted genes (or about 1500 genes total) [22]. This numerical discrepancy makes it rather unlikely that a single snoRNA is responsible for the regulation of all ESRE genes, unless the snoRNA were to recognize the ESRE consensus sequence. Sequence comparisons suggest that none of the snoRNAs predicted in the *C. elegans* genome are obvious candidates for recognizing the ESRE motif. In all cases we are aware of, the hybridization sequence for snoRNAs is located between the C and D boxes, and no *C. elegans* box C/D snoRNA matches the ESRE consensus site.

An alternative hypothesis is that stress signals change snoRNP expression or targeting, changing rRNA modification patterns since most known box C/D snoRNP targets are in rRNA. Recent observations have shown that some of these sites are not saturated under normal growth conditions and that the degree to which some sites are modified has been linked with cellular stress [78]. This has led to the intriguing suggestion that different populations of ribosomes, perhaps with specific functions, may exist within cells. If this were the case, it would be reasonable to expect that one sub-population has specialized for stress-response genes, allowing rapid and dynamic switching of the translational profile of the cell to support survival. Indeed, a previous report has showed that ribosome methylation can regulate translation [79]. This is functionally analogous to the well-known integrated stress response, where

phosphorylation of eIF-2α substantially reduces cap-dependent mRNA translation in favor of translation from structured mRNA elements called internal ribosomal entry sites, or IRESes [16]. snoRNA-mediated changes to the ribosome could similarly facilitate specialized translation of structured mRNAs.

If this were the case, one expected outcome would be increased ESRE gene expression during stress. But while we do see increased expression of ESRE genes during stress, this is at least partially due to transcriptional changes that we have measured using qRT-PCR; in contrast, we have not seen evidence of translational differences. Additionally, ESRE genes are not known to be activated by translational inhibitors like *P. aeruginosa* exotoxin A or hygromycin [1]. Targeting the 48S pre-initiation complex here (**Fig 5A**) also did not activate ESRE gene regulation. We also saw no changes in ESRE gene expression when components of the box H/ACA snoRNP complex were disrupted. rRNA modification by these ribonucleoprotein complexes is also important and would also be expected to affect transcription if this were a mechanism of ESRE gene regulation. Finally, stress-responsive ribosomal modifications wouldn't explain the association of the box C/D snoRNP complexes with the ESRE motif. Ultimately, the data do not currently support this model.

Importantly, we found that immune effectors, such as *irg-1* (**S6A Fig** and [42]) and *irg-5* (**Fig 6**), were upregulated by the absence of box C/D snoRNPs. This is consistent with many reports that disruption of core cellular processes activates innate immune pathways [1,9,33]. In this case, the loss of box C/D snoRNPs activated *irg-1* via translation suppression [42] and *irg-5* through an unknown mechanism that may be related to mitochondrial dysfunction. Consistent with this, we saw increased host survival after box C/D snoRNP knockdown in a pathogenesis assay that depends on intestinal colonization. In contrast, the loss of the box C/D snoRNPs impaired worm survival in a pathogenesis setting where mitochondria are severely compromised (the Liquid Killing assay [58]), which may result from the combined mitochondrial disruption of box C/D loss and pathogen damage.

Interestingly, although the loss of ATF-7/ATF7 or PMK-1/p38 MAPK was able to completely abolish induction of *irg-5* and several other immune genes after RNAi-mediated upregulation, the loss of box C/D snoRNPs still increased survival in *atf-7(gk715)* and *pmk-1 (km25)* mutants. This suggests a different biological significance of the box C/D snoRNPs in modulating innate immunity and mitochondrial surveillance.

These data lead to a model where the box C/D snoRNPs act as a molecular switch that activates quality control pathways while inhibiting immune responses (**Fig 10**). Our data showed that box C/D snoRNPs are necessary for normal function of ESRE and UPR$^{mt}$ (and ATFS-1) and their disruption leads to increased basal *Ptbb-6*::GFP expression, consistent with previous findings where we and others [18,21] have showed that ATFS-1 may act to limit MAPK$^{mt}$ expression under normal conditions but works in concert to help restore cellular health under stress. The purpose of this novel mechanism may be to allow mitochondria an opportunity to repair before cellular defenses (which require larger energy expenditure) are activated. Future work will focus on understanding the relationships between these systems.

## Materials and methods

### *C. elegans* strains

All *C. elegans* strains were maintained on standard nematode growth medium (NGM) [80] seeded with *Escherichia coli* strain OP50 as a food source and were maintained at 20˚C [80], unless otherwise noted. *C. elegans* strains used in this study included N2 Bristol (wild-type), SS104 [*glp-4(bn2)*], COP262 |*knuSi221* [*Pfib-1*::FIB-1(genomic)::eGFP::*fib-1* 3' UTR + *unc-119* (+)]| [81], WY703 |*fdIs2* [*3XESRE*::GFP; *pFF4*[*rol-6(su1006)*]| [27], SJ4100 |*zcIs13* [*Phsp-6*::

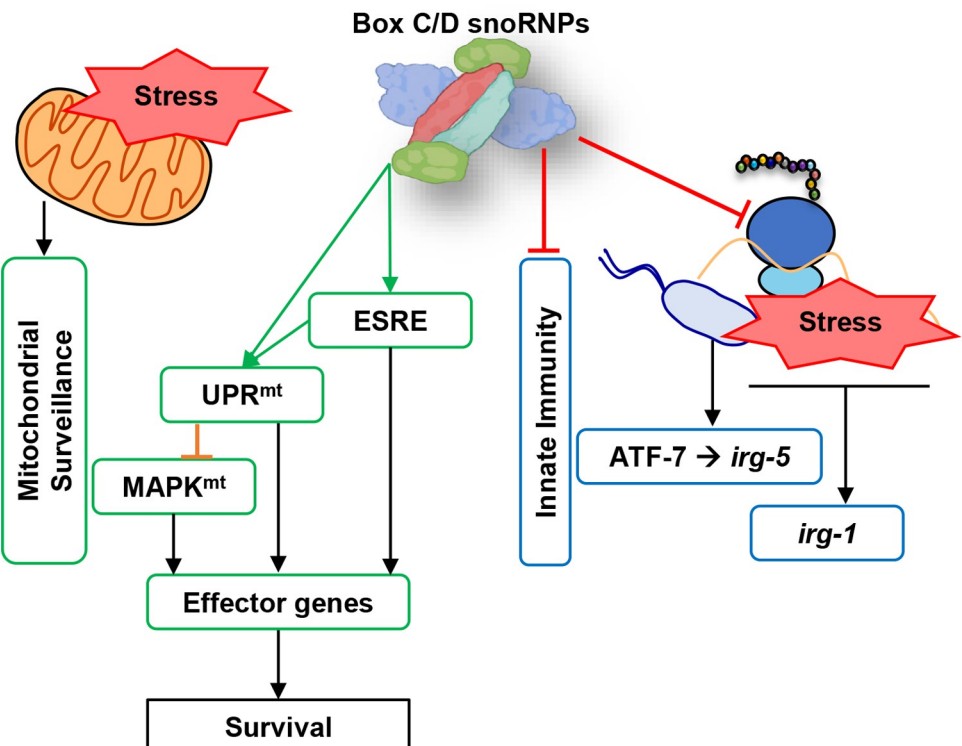

**Fig 10. Proposed model of box C/D snoRNP roles in cellular pathways regulation.** Box C/D snoRNPs act as a molecular switch that suppresses innate immunity and activates mitochondrial pathways upon stress.

GFP]|,|*atfs-1*(*et15*); *zcIs9* [*Phsp-60*::*GFP*]| [82], NVK235 (*zcIs13*; *Patfs-1ΔESRE*::ATFS-1^WT) [21], SLR115 |*dvIs67* [*Ptbb-6*::GFP + *Pmyo-3*::dsRed]| [18], AY101 |*acIs101* [*pDB09.1*(*Pirg-5*:: GFP); *pRF4*[*rol-6*(*su1006*)]]| [54], AU133 |*agIs17*[*Pmyo-2*::mCherry + *Pirg-1*::GFP]| [2], AU078 [*PT24B8.5*::GFP::*unc-54* 3' UTR], VC1518 [*atf-7*(*gk715*)], and KU25 [*pmk-1*(*km25*)]. Worms were synchronized by hypochlorite isolation of eggs from gravid adults, followed by hatching of eggs in S Basal. 6,000 synchronized L1 larvae were transferred onto 10 cm NGM plates seeded with OP50. After transfer, worms were grown at 20°C for 46 hours prior to experiments, or for three days for production of the next cycle of eggs. Young adult hermaphrodites were used for all assays.

## Bacterial strains

RNAi experiments in this study were done using RNAi-competent *E. coli* HT115 obtained from the Ahringer or Vidal RNAi libraries [83,84]. Luciferase (L4440::Luc) RNAi-expressing *E. coli* was obtained from the Antebi Lab [85]. All RNAi plasmids were sequence-verified prior to use. *P. aeruginosa* strain PA14 was used for all infectious assays [86].

## RNA interference protocol

RNAi-expressing *E. coli* were cultured and seeded onto NGM plates supplemented with 25 μg/ mL carbenicillin and 1 mM IPTG. When double RNAi was performed, bacterial cultures were mixed in a 1:1 ratio. For RNAi experiments starting at L1, 2,000 synchronized L1 larvae were transferred onto 6 cm RNAi plates and grown at 20°C for 46 hours prior to imaging or exposure to chemical compounds or pathogens. For RNAi experiment starting at L3, 2,500

synchronized L1 larvae were transferred onto 6 cm regular NGM plates seeded with OP50 and grown at 20˚C for 22 hours until reaching the L3 stage. Worms were then washed off plates, rinsed three times, and transferred onto RNAi plates. Worms were grown at 20˚C for 24 hours on RNAi plates prior to use for experiments.

### Biochemical assays

Synchronized young adult worms were exposed to 1 mM 1,1-phenanthroline for 20 hours, 50 μM rotenone for 14 hours, or DMSO (solvent control). After exposure, worms were collected in 15 mL conical tubes and washed three times to remove residual compounds, and snap frozen in liquid nitrogen. Frozen worm pellets were homogenized using a Dounce homogenizer with the appropriate volume of Cytoplasmic Extraction Reagent I (NE-PER Nuclear and Cytoplasmic Extraction Reagents, Thermo Scientific). Cytoplasmic and nuclear protein extraction was then performed according to the manufacturer's protocol. Extracts were immediately used for electrophoretic mobility shift assay (EMSA) and oligo pulldown, otherwise were stored at 4˚C (maximum for a week).

EMSA was performed by incubating worms' nuclear-enriched extract for 20 minutes at room temperature with biotinylated oligos (4XESRE or mutated 4XESRE) (ordered from IDT) as bait in Binding Buffer (50 ng/μL poly (dI•dC), 5% glycerol, 0.1% NP-40, 2.5 mM MgCl2, 1 mM EDTA, and 20 fmol biotinylated oligos). Binding reactions were then loaded for on a polyacrylamide gel and run until the dye front had migrated ¾ down the length of the gel. Material was then transferred to a nylon membrane for 30 minutes at 380 mA. DNA on the membrane was then crosslinked at 120 mJ/cm2 using ultraviolet light. Biotin-labeled DNA was then detected using the LightShift Chemiluminescent EMSA Kit (Thermo Scientific) and exposed to X-ray film.

Oligo pulldown was performed according to the manual for DynaBeads M-280 Streptavidin (Invitrogen). In short, magnetic beads were first washed with Binding and Washing buffer (BW 2X) (10 mM Tris-HCl pH 7.5, 1 mM EDTA, and 2 M NaCl). Beads were then coupled with the biotinylated-4XESRE oligos bait for 15 minutes (with BW 1X). Coated beads were resuspended in PBS buffer (0.1 M phosphate, 0.15 M NaCl) pH 7.4 and then incubated with worm nuclear extract for 2 hours at room temperature, washed, and eluted. Elution samples were sent for tandem MS/MS.

### *C. elegans* size measurement

Synchronized young adult *glp-4(bn2)* worms were washed from NGM plates seeded with RNAi-expressing *E. coli* HT115 into a 15 mL conical tube and rinsed three times. Worm size was determined via flow vermimetry by using COPAS Biosort machine (Union Biometrica).

### *C. elegans* lifespan assay

50 young adult *glp-4(bn2)* worms were transferred onto NGM plates seeded with *E. coli* OP50 and incubated at 25˚C. Worms were scored daily to obtain survival curves; dead worms were removed from assay plates. Three technical replicates (total: 150 worms per condition) in each of the three biological replicates were performed. Worms that left the surface of the plates were censored from analysis.

### *C. elegans* chemical exposure assays

Synchronized young adult worms were washed from NGM plates seeded with *E. coli* OP50 or with RNAi-expressing *E. coli* HT115 into 15 mL conical tubes and rinsed three times. Worms

were then sorted into a 96-well plate (~100 worms/well). S Basal supplemented with 50 μM rotenone (Sigma), 2 mg/mL cycloheximide (Sigma), or DMSO (solvent control) was then added into the wells of the 96-well plate to a final volume of 100 μL. Worms were imaged with Cytation5 automated microscope every two hours for 20 hours at room temperature [87]. At least three biological replicates were performed for each experiment.

## Quantitative reverse transcriptase PCR (qRT-PCR)

8,000 young adult worms were used for RNA extraction, and subsequent qRT-PCR was performed as previously described [58]. Prior to RNA extraction, worms were reared on RNAi plates starting at the L1 or L3 stage, as described above. For treatment with rotenone, young adult worms were washed off plates, rinsed three times, and were incubated for 8 hours in S Basal supplemented with 25 μM rotenone or a corresponding volume of DMSO control. For each experiment, at least three biological replicates were performed. Primer sequences are listed on **S5 Table**.

## *C. elegans* pathogenesis assays

*P. aeruginosa* liquid pathogenesis model (Liquid Killing or LK) was performed essentially as described [58,88]. 25 synchronized young adult worms were sorted into each well of a 384-well plate. Liquid Killing medium was mixed with *P. aeruginosa* PA14 (final $OD_{600}$: 0.03) and then added into each well. Plates were incubated at 25˚C. At varying time points, plates were washed three times and worms were stained with SYTOX Orange nucleic acid stain (Invitrogen) for 12 hours to identify dead worms. Plates were then washed and imaged with a Cytation5 automated microscope and the fraction of dead worms was autonomously quantified with CellProfiler.

   *P. aeruginosa* agar pathogenesis model (Slow Killing or SK) was performed as previously described [89]. 50 young adult worms were transferred onto PA14-SK plates and incubated at 25˚C. Worms were scored daily to obtain survival curves; dead worms were removed from assay plates and worms that left the surface of the plate were censored from data analysis.

   *P. aeruginosa* exposure to worms carrying *Pirg-5*::GFP (AY101) was performed similarly as SK assay. 500 worms were transferred onto PA14-SK plates. After 8 hours, worms were washed off plates and transferred into a 96-well plate and washed five times to remove bacteria. Imaging and GFP quantification were performed with Cytation5 automated microscope and Gen5 3.0 software.

## Colony forming unit (CFU) assay

18 worms were collected from a Slow Killing agar plate after 48 hours of exposure. Worms were washed three times to remove residual bacteria and the volume was aspirated to 100 μL. 100 μL of carbide beads were added, and tubes were vortexed vigorously to break open the worms. Supernatant was serially diluted and plated on LB plates, incubated overnight, and colonies were counted the next day. Assays were performed in triplicate.

## Microscopy

For visualization of the worm reporter strain COP262 (*Pfib-1*::FIB-1::eGFP), worms were immobilized using 1 mM levamisole and then transferred onto a 3% noble agar pad. Three biological replicates with at least 20 worms per replicate were imaged using fluorescence microscope (Zeiss ApoTome.2 Imager.M2, Carl Zeiss, Germany) with 63x magnification. *Pfib-1*::FIB-1::eGFP fluorescence in the intestine of each animal was imaged and quantified.

## Fluorescence quantification

For visualization of the worm reporter strains AU078, AU133, AY101, NVK235, SJ4100, SLR115, and WY703 in 96-well plates, Cytation5 Cell Imaging Multi-Mode Reader (BioTek Instruments) was used. All imaging experiments were performed with identical settings between replicates of each assay. Imaging settings were optimized depending on the strength of the reporter's signal. GFP quantification was performed by using Gen5 3.0 software and/or via flow vermimetry (COPAS Biosort machine from Union Biometrica) [87].

For fluorescence quantification of the worm reporter strain COP262, flow vermimetry (COPAS Biosort machine, Union Biometrica) was used. For quantification of the area of FIB-1::eGFP fluorescence, ImageJ [90] was used. In short, images were converted to 16-bit and threshold was adjusted to identify fluorescent regions. Particles with the minimum size of 700 pixels$^2$ were then selected throughout the image and the area was calculated by the software.

## Mitochondrial membrane potential and mitochondrial mass quantification

Approximately 400 N2 worms were washed off RNAi plates and transferred to 4 wells of a 96-well plate (~100 worms per well). Fluorescent dye (4.375 μM of MitoTracker Red or 4.375 μM of MitoTracker Green) or an appropriate volume of DMSO control was then added into each well. After 1 hour of incubation, worms were washed three times to remove any remaining dye. Fluorescence measurements were taken via flow vermimetry (COPAS Biosort, Union Biometrica). At least three biological replicates were performed.

## Oxygen consumption rate measurement

6,000 N2 worms per condition (reared on *E. coli* expressing empty vector or targeting box C/D snoRNPs RNAi; untreated or treated with DMSO or 25 μM rotenone for 8 hours) were pipetted into the biological oxygen monitor instrument (YSI 5300). Oxygen consumption was measured for ten minutes by using the monitor with a Clark-type oxygen electrode (YSI 5301, Yellow Springs Instrument) at 20°C as previously described [91]. At least three biological replicates were performed.

## Statistical analysis

All numerical data underlying all graphs can be found in **S1 Data**. RStudio (version 3.6.3) was used to perform statistical analysis. One-way or two-way analysis of variance (ANOVA) were performed to calculate the significance of a condition or treatment when there were three or more groups in the experimental setting. The statistically significant experimental results as determined via ANOVA were then followed by Dunnett's test (R package DescTools, version 0.99.34) to calculate statistical significance or *p* values between each group compared to the control group. Student's *t*-test analyses were performed to calculate the *p* values when comparing two groups in an experimental setting. A Log-rank test (http://bioinf.wehi.edu.au/software/russell/logrank/) was performed to calculate the *p* values in the lifespan and Slow Killing assays. Dunnett's test, Student's *t*-test, and log-rank test results were indicated in graphs as follows: NS not significant, $^*p < 0.05$, $^{**}p < 0.01$, and $^{***}p < 0.001$.

## Supporting information

**S1 Fig. Proteomic assays revealed the presence of ESRE-binding factor(s).** Electrophoretic mobility shift assay (EMSA) showed the presence of the ESRE-binding motif through the identified 'Shift'. Worms were treated with DMSO or 50 μM rotenone for 14 hours. D: DMSO-

treated, R: rotenone-treated, 4xESRE: oligo bait, and 4xESRE(M): oligo bait with mutations. (TIFF)

**S2 Fig. The use of RNAi targeting luciferase yielded in similar results of multiple reporters' expressions as compared to empty vector control in both single and double RNAi settings.** **(A)** Quantification of GFP fluorescence of *C. elegans* carrying *3XESRE*::GFP that were reared on *E. coli* expressing RNAi targeting empty vector (*EV*) or *luciferase*. Worms were treated for 8 hours with vehicle (DMSO) or 25 μM rotenone. RNAi was started at L1 (left) or L3 stage (right). **(B)** Quantification of GFP fluorescence of *C. elegans* carrying *Phsp-6*::GFP (top) and *Ptbb-6*::GFP (bottom) reporters that were reared on *E. coli* expressing RNAi targeting empty vector (*EV*) or *luciferase*. Double RNAi was performed with empty vector (*EV*) or *luciferase*, or *spg-7(RNAi)*. Three biological replicates with ~400 worms/replicate were analyzed. *p*-values were determined from Student's *t*-test. GFP values were normalized to *EV*-DMSO or *EV*. NS not significant, *$p < 0.05$, ** $p < 0.01$, *** $p < 0.001$. In **(A)**, purple significance marks indicate comparison of *luciferase(RNAi)* to *EV(RNAi)* in DMSO, red marks indicate comparison of *luciferase(RNAi)* to *EV(RNAi)* in rotenone, and black marks indicate comparison between expressions in rotenone and DMSO. In **(B)**, red significance marks indicate comparison between *luciferase(RNAi)* to the corresponding condition but with *EV(RNAi)*, e.g., *fib-1;luciferase(RNAi)* vs. *fib-1;EV(RNAi)* and *spg-7;luciferase(RNAi)* vs *spg-7;EV(RNAi)*. (TIF)

**S3 Fig. Knockdown of box C/D snoRNP assembly factor RUVB-1 slightly affected mitochondrial surveillance pathways.** Quantification of GFP fluorescence of *C. elegans* carrying **(A)** *3XESRE*::GFP, **(B)** *Phsp-6*::GFP, and **(C)** *Ptbb-6*::GFP reporters that were reared on *E. coli* expressing RNAi targeting empty vector (*EV*) or *ruvb-1/RUVB*. In **(A)**, worms were treated for 8 hours with vehicle (DMSO) or 50 μM rotenone. In **(B, C)**, double RNAi was performed with empty vector (*EV*) or *spg-7(RNAi)*. **(D)** Images and fluorescence intensity quantification of *C. elegans* carrying *Pfib-1*::FIB-1::eGFP reporter that were reared on *E. coli* expressing RNAi targeting empty vector (*EV*), *fib-1/FBL*, or *ruvb-1/RUVB*. RNAi was started at L1 or L3 stage as indicated in the figure. Three biological replicates with **(A-C)** ~400 worms/replicate or **(D)** ~25 worms/replicate were analyzed. *p*-values were determined from Student's *t*-test. GFP values were normalized to *EV*-DMSO or *EV*. NS not significant, *$p < 0.05$, ** $p < 0.01$, *** $p < 0.001$. In **(A-C)**, purple significance marks indicate comparison of *ruvb-1(RNAi)* to *EV(RNAi)* in unstressed condition (DMSO or *EV(RNAi)*), red marks indicate comparison of *ruvb-1(RNAi)* to *EV(RNAi)* in stressed condition (rotenone or *spg-7(RNAi)*), and black marks indicate comparison between stressed and unstressed conditions. (TIFF)

**S4 Fig. RNAi targeting core members of box H/ACA snoRNPs did not affect mitochondrial surveillance pathways.** Quantification of GFP fluorescence of *C. elegans* carrying **(A)** *3XESRE*::GFP, **(B)** *Phsp-6*::GFP, or **(C)** *Ptbb-6*::GFP reporters that were reared on *E. coli* expressing empty vector (*EV*) or RNAi targeting box H/ACA snoRNP members: *nola-3/Nop10*, *Y48A6B.3/Nhp2*, or *Y66H1A.4/Gar1*. In **(A)**, worms were treated for 8 hours with vehicle (DMSO) or 50 μM rotenone. In **(B, C)**, double RNAi was performed with empty vector (*EV*) or *spg-7(RNAi)*. Three biological replicates with ~400 worms/replicate were analyzed. *p*-values were determined from two-way ANOVA, followed by Dunnett's test, and Student's *t*-test. All fold changes were normalized to DMSO-*EV* or *EV* control. NS not significant, *$p < 0.05$, ** $p < 0.01$, *** $p < 0.001$. In all panels, purple significance marks indicate comparison of *H/ACA genes(RNAi)* to *EV(RNAi)* in unstressed condition (DMSO or *EV(RNAi)*), red marks indicate comparison of *H/ACA genes(RNAi)* to *EV(RNAi)* in stressed condition

(rotenone or *spg-7(RNAi)*), and black marks indicate comparison between stressed and unstressed conditions.
(TIF)

**S5 Fig. Quantitative RT-PCR verified reduction of transcripts in RNAi targeting members of box H/ACA and eukaryotic 48S transcription initiation complex.** mRNA levels of target genes in N2 worms reared on *E. coli* expressing RNAi targeting *Y48A6B.3*, *clu-1/eIF3A*, *ife-2/eIF4E*, or *inf-1/eIF4A*. Three biological replicates with ~8,000 worms/replicate were analyzed. All fold changes were normalized to expression of target genes in worms reared on *EV(RNAi)*-expressing *E. coli*.
(TIF)

**S6 Fig. Knockdown of box C/D snoRNPs induced immune response gene *irg-1*. (A, B)** Quantification of GFP fluorescence of *C. elegans* carrying *Pirg-1*::GFP reporter that **(A)** were reared on *E. coli* expressing empty vector (*EV*), *fib-1(RNAi)*, *nol-56(RNAi)*, or *nol-58(RNAi)* or **(B)** exposed to 2 mg/mL cycloheximide or DMSO control for 8 hours. Three biological replicates with ~400 worms/replicate were analyzed. *p*-values were determined from **(A)** one-way ANOVA, followed by Dunnett's test or **(B)** Student's *t*-test. All fold changes were normalized to *EV* or DMSO control. $^{*}p < 0.05$, $^{**}p < 0.01$.
(TIF)

**S7 Fig. Cycloheximide treatment did not affect mitochondrial surveillance pathways.** Quantification of GFP fluorescence of *C. elegans* carrying **(A)** *3XESRE*::GFP or **(B)** *Ptbb-6*::GFP reporters that were treated for 8 hours with vehicle (DMSO) or 2 mg/mL of translation elongation inhibitor cycloheximide. Three biological replicates with ~400 worms/replicate were analyzed. *p*-values were determined from Student's *t*-test. NS not significant.
(TIF)

**S8 Fig. RNAi targeting core members of box H/ACA snoRNPs did not affect innate immune pathways.** Quantification of GFP fluorescence of *C. elegans* carrying **(A)** *Pirg-1*::GFP or **(B)** *Pirg-5*::GFP reporters that were reared on *E. coli* expressing empty vector (*EV*) or RNAi targeting box H/ACA snoRNP members: *nola-3/Nop10*, *Y48A6B.3/Nhp2*, or *Y66H1A.4/Gar1*. Three biological replicates with ~400 worms/replicate were analyzed. *p*-values were determined from one-way ANOVA, followed by Dunnett's test. All fold changes were normalized to *EV* control. NS not significant.
(TIF)

**S9 Fig. RNAi targeting box C/D snoRNPs provides protection in *atf-7(gk715)* and *pmk-1 (km25)* mutants in agar pathogenesis model. (A, B)** Survival of N2 wild-type, *atf-7(gk715)*, or *pmk-1(km25)* mutants reared on RNAi strains targeting empty vector (*EV*) and **(A)** *nol-56* or **(B)** *nol-58*. Three biological replicates with ~150 worms/replicate were analyzed. Representative replicates are shown. *p*-values were determined from log-rank test. $^{***}p < 0.001$. Purple significance marks indicate comparison of *atf-7(gk715)* or *pmk-1(km25)* mutants to N2 wild-type (reared on *EV(RNAi)*), red marks indicate comparison of *atf-7(gk715)* or *pmk-1(km25)* mutants to N2 wild-type (reared on *nol-56(RNAi)* or *nol-58(RNAi)*), and black marks indicate comparison between *nol-56(RNAi)* or *nol-58(RNAi)* vs. *EV(RNAi)* for each worm strain.
(TIF)

**S1 Table. List of 75 protein candidates obtained from mass spectrometry.**
(XLSX)

**S2 Table. Raw data of lifespan measurement of *glp-4(bn2)* worms.** Animals were reared on RNAi targeting box C/D snoRNPs or empty vector control. Raw data of 3 replicates were shown.
(XLSX)

**S3 Table. Raw data of survival of *glp-4(bn2)* worms under Slow Killing assay.** Animals were reared on RNAi targeting box C/D snoRNPs or empty vector control. Raw data of 3 replicates were shown.
(XLSX)

**S4 Table. Raw data of survival of N2 wild-type and *atf-7(gk715)* and *pmk-1(km25)* mutants under Slow Killing assay.** Worms were reared on RNAi targeting box C/D snoRNPs or empty vector control. Raw data of 3 replicates were shown.
(XLSX)

**S5 Table. Sequences of forward and reverse primers for quantitative RT-PCR experiments used in this study.**
(XLSX)

**S1 Data. Numerical data underlying all graphs.**
(XLSX)

## Acknowledgments

*C. elegans* strains used were provided by David Fay, Cole Haynes, or obtained from the CGC, which is funded by NIH Office of Research Infrastructure Programs (P40 OD010440). *E. coli* expressing *luciferase(RNAi)* was provided by the laboratory of Adam Antebi.

## Author Contributions

**Conceptualization:** Elissa Tjahjono, Natalia V. Kirienko.

**Formal analysis:** Elissa Tjahjono, Alexey V. Revtovich, Natalia V. Kirienko.

**Funding acquisition:** Natalia V. Kirienko.

**Investigation:** Elissa Tjahjono, Alexey V. Revtovich.

**Supervision:** Natalia V. Kirienko.

**Visualization:** Elissa Tjahjono, Alexey V. Revtovich, Natalia V. Kirienko.

**Writing – original draft:** Elissa Tjahjono, Natalia V. Kirienko.

**Writing – review & editing:** Elissa Tjahjono, Alexey V. Revtovich, Natalia V. Kirienko.

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
