## [Decision Letter · Decision Letter 0]

8 Nov 2021

Dear Dr Kirienko,

Thank you very much for submitting your Research Article entitled 'Box C/D Small Nucleolar Ribonucleoproteins Regulate Mitochondrial Surveillance and Innate Immunity' to PLOS Genetics.

The manuscript was fully evaluated at the editorial level and by independent peer reviewers. The reviewers appreciated the attention to an important problem, but raised some substantial concerns about the current manuscript. Based on the reviews, we will not be able to accept this version of the manuscript, but we would be willing to review a much-revised version. We cannot, of course, promise publication at that time.

If you decide to revise the manuscript for further consideration at PLOS Genetics, please aim to resubmit within the next 60 days, unless it will take extra time to address the concerns of the reviewers, in which case we would appreciate an expected resubmission date by email to plosgenetics@plos.org.

[LINK]

We are sorry that we cannot be more positive about your manuscript at this stage. Please do not hesitate to contact us if you have any concerns or questions.

Yours sincerely,

Javier E. Irazoqui

Associate Editor

PLOS Genetics

Gregory P. Copenhaver

Editor-in-Chief

PLOS Genetics

Reviewer's Responses to Questions

**Comments to the Authors:**

Reviewer #1: The manuscript by Tjahjono, Revtovich, and Kirienko describe a new role for the box C/D small nucleolar ribonucleoproteins in regulating mitochondrial recovery and innate immunity. The authors wished to identify new regulators of the ESRE network, a mitochondrial surveillance pathway that comprises genes whose promoters contain the ESRE motif. Using a biochemical pulldown approach involving a tandem ESRE motif repeat oligonucleotide, the authors identified the box C/D snoRNP complex members FIB-1 and NOL-56, as regulators of the ESRE pathway. In addition, box C/D snoRNPs also regulated other mitochondrial surveillance pathways including the UPRmt and MAPKmt. The spatial requirement of the snoRNP in regulating these mitochondrial surveillance pathways was also examined using RNAi against ruvb-1, a component required for assembly and localization of box C/D snoRNP to the nucleoli. Nucleolar localization of snoRNP was required for ESRE regulation but only partially for the MAPKmt pathway and was not required for the UPRmt. The authors also confirmed that members of the H/ACA snoRNP complex do not have a role in mitochondrial surveillance and furthermore, that defects in translation are unlikely the cause for the effects observed with box C/D snoRNP knockdown. Lastly, using previously established reporters, the authors find that box C/D snoRNP functions to repress the innate immune response. Consistent with this finding, knockdown of members of the box C/D snoRNP increases host survival during infection with P. aeruginosa under slow-killing conditions in which the host succumbs to pathogen colonization. However, loss of box C/D snoRNP members rendered animals more sensitive to P. aeruginosa under liquid-killing conditions, which kills the host via iron sequestration from mitochondria.

The manuscript is overall a well-written body of work and the identification of the box C/D snoRNP complex as a regulator of mitochondrial surveillance and host immunity is novel and will be of interest to those studying cellular stress response pathways and host-pathogen interactions. However, the following suggestions could help improve the author’s findings.

Major revisions

The authors’ model suggests that box C/D snoRNPs encourage mitochondrial surveillance pathways by suppressing the immune response which can be quite energy-consuming. However, these conclusions are based on a handful of specific gene transcriptional GFP reporters and limited survival/lifespan assays. I would suggest the following to make their model more substantive.

1. The authors suggest that loss of box C/D snoRNPs reduces mitochondrial recovery, a finding that is based on altered activity of mitochondrial surveillance reporters. The authors should assay mitochondrial functions in the absence of box C/D snoRNPs such as mitochondrial membrane potential and/or oxygen consumption rates.

2. It is intriguing that loss of fib-1, but not other box C/D snoRNP complex members, results in increased lifespan (Figure 1E) since mitochondrial surveillance is predicted to be compromised in this animal. Mitochondrial stress is a known inducer of increased longevity, as observed with mitochondrial mutants clk-1, isp-1, nuo-6 or cco-1 RNAi, among others. The UPRmt and MAPKmt are also required for the extended lifespan of some of these conditions. It would be interesting to measure the lifespan for some of these long-lived mitochondrial stressed animals in the presence or absence of FIB-1. If FIB-1 is truly promoting mitochondrial surveillance, then the increased lifespan of fib-1 RNAi animals should not be additive with the increased lifespan of these mitochondrial stressed animals. Instead, fib-1 RNAi should reduce the lifespans of these animals (possibly to levels observed with fib-1 RNAi alone) based on the authors’ model, due to their compromised mitochondrial surveillance ability.

3. Related to comment 2) the authors did not observe increased lifespan in the absence of nol-56/58. Is this a consequence of stronger RNAi knockdown of these genes compared to fib-1 RNAi? If nol-56/58 RNAi were to be diluted, would they observe an increase in lifespan or is this effect specific to fib-1 RNAi?

4. a) The author’s model proposes that the box C/D snoRNPs repress immunity to favor mitochondrial surveillance. They show that two immune reporters, irg-1::GFP and irg-5::GFP are induced with box C/D snoRNP RNAi. Ideally, a more global view of transcriptional changes occurring in the presence or absence of box C/D snoRNP using RNAseq would provide more support for their model. Alternatively, one could test by qPCR a set of immune genes that are known to be regulated by mitochondrial surveillance programs (e.g. Campos et al. 2021 PMID 34617666, Pellegrino et al. 2014; PMID 25274306) in the presence or absence of box C/D snoRNPs.

b) Also, pathogen gut colonization should be tested to complement their slow-killing survival assay and to further support their model that immune responses and host resistance are enhanced with loss of box C/D snoRNP.

5. The authors find that transcriptional induction of immune genes following loss of box C/D snoRNP is partially or fully dependent on PMK-1/p38 and ATF-7/ATF7. The authors should perform their slow-killing survival assays in the presence or absence of PMK-1/ATF-7 to support their gene expression data.

Also, is ZIP-2 implicated in the increased survival of box C/D snoRNP deficient animals? This seems relevant since irg-1 was induced in the absence of box C/D snoRNP and ZIP-2 is a critical regulator of this gene during P. aeruginosa infection and with translation repression. As well, the authors previously determined ZIP-2 to be related to ESRE gene expression regulation.

Minor revisions

1. lines 201-208: The authors should speculate why the localization of box C/D snoRNPs is relevant for the regulation of some, but not all, of the mitochondrial surveillance pathways.

2. lines 214-16: The authors should comment further on their finding of increased FIB-1::eGFP punctae size with rotenone. What is the significance of this effect? Also, is the increased size of FIB-1::eGFP punctae specific for rotenone treatment? Do other mitochondrial stress conditions that activate the UPRmt/MAPKmt have a similar effect? Do they also observe increase puncta size during infection with P. aeruginosa slow-killing/liquid killing conditions?

3. More details on statistical analysis for lifespan and survival assays are needed. A supplementary table listing all survival/lifespan replicates should be provided. Included in this table should be the number of animals censored for their experiments. Also, statistical comparisons for these assays should follow the guidelines found in Petrascheck and Miller (2017); PMID 28713422.

4. The authors should provide more details in the figure legends of the color codes used for their statistical comparisons and what they signify.

5. Figure 2C: the pictures of tbb-6::GFP in the absence of stress do not obviously reflect the increase in expression that is reported in the quantification graph for fib-1, nol-56, and nol-58 RNAi. Perhaps a higher exposure would allow a better distinction between treatments?

6. Typo on line 160: “we triggered activation of the of downstream effectors for”. Remove “of”.

7. Typo on line 292-3: “its known regulation by the ATF-7/ATF7” should read “its known regulation by the transcription factor ATF-7/ATF7”.

Reviewer #2: Mitochondrial surveillance is crucial for maintaining organismal health under various stress conditions, including pathogen infections. In this manuscript entitled “Box C/D Small Nucleolar Ribonucleoproteins Regulate Mitochondrial Surveillance and Innate Immunity”, the authors reported novel functions of the box C/D snoRNA core proteins (snoRNPs) in upregulating mitochondrial surveillance and modulating immune responses. The authors previously reported the roles of Ethanol and Stress Response (ESRE) pathway and ESRE motif in mitochondrial surveillance in response to intracellular stressors. In the current work, the authors found that the box C/D snRNP component proteins, including FIB-1, NOL-56, and NOL-58, upregulated the ESRE pathway and mitochondrial unfolded protein response (UPRmt) under mitochondrial stress conditions. The authors identified the box C/D snRNP component proteins from ESRE motif-binding proteins by using pulldown and mass spec analysis. They showed that knockdown of the box C/D snRNP component proteins upregulated MAPKmt stress pathways. In addition, knocking down the box C/D snoRNP components upregulated immune effectors and altered the resistance of C. elegans against pathogenic bacteria. Overall, the authors suggested that box C/D snoRNPs act as molecular switch between mitochondrial surveillance and innate immunity through ESRE and MAPKmt signaling pathways. This paper starts with the unbiased biochemical identification for ESRE-regulating factors in combination with RNAi screen, and ends with molecular genetic analysis for functional importance of the the box C/D snRNP for immunity. This work is thorough and novel, and will provide valuable information regarding the research field of mitochondrial biology and immunity. I have basically one major concern in addition to minor ones that will further improve the quality of this excellent paper.

Major comments

1. The genetics in this paper solely depends on RNAi knockdown. RNAi experiments are variable and less reliable than those with mutants, in particular when the data are negative and double RNAi is used. I strongly recommend that the authors strengthen some of the data by performing additional experiments and also by discussing the caveat of using RNAi in the Discussion. Following are my specific suggestions.

1.1. Double RNAi experiments in Figures 2 and 3. The authors performed double RNAi experiments using RNAi-expressing bacteria mixture. They need to show the efficiency of RNAi for the experiments using qRT-PCR (or less preferentially fluorescence reporters) to rule out the possibility that RNAi clones work less efficiently.

1.2. Negative data with RNAi in Figures 4, S4 and S7. Here we don’t know whether RNAi simply did not work or work less efficiently for these experiments. The authors again need to perform qRT-PCR or reporter assays.

1.3. Additional experiments with pmk-1 and atf-7 mutants for Figures 5 and 6. As the mutants are available for pmk-1 and atf-7, and some of the data are the key data, I think they need to test whether the partial and complete requirement of pmk-1 and atf-7 are caused by hypomorphic nature or RNAi.

1.4. Discussion for the limitation of RNAi. Validating RNAi data with the above experiments will dramatically improve the quality of the paper, but practically I also understand that it will require enormous time and efforts. I think the authors may perform some key experiments for validation of RNAi and discuss the limitation of remaining data in the Discussion.

Minor comments

1. The authors showed the quantification data of GFP fluorescence with representative fluorescent images in Figures 1 and 2. However, they did not display fluorescent images for Figures 3 to 6. It will be better to show fluorescent images of following strains that are not shown in Figures 1 and 2: Phsp-6::GFP reporter strains that were crossed with Patfs-1ΔESRE::ATFS-1WT in Figure 3A or Pirg-5::GFP reporter treated with fib-1, nol-56, or nol-58 RNAi in Figure 5.

2. The central part of the Abstract should be re-written with a little more details to deliver the main results of this paper better.

3. The authors explained three mitochondrial surveillance pathways based on previous studies with various organisms. Those organisms need to be specified in the Introduction.

4. On page 6, the authors identified regulatory components of the ESRE pathway by using pulldown assays. It will be better for the authors to add a Supplemental table for listing up those 75 candidates that were identified.

5. In Figure legends, please explain specific information regarding the data with more details. For example, the authors did not explain the meaning of the colors of in the bar graphs for statistical significance at the beginning, although they explained this on page 9.

6. On page 9, in figure 3, they may want to move Figure S1B to Figure 3B for better comparison between the effects of RNAi starting at L3 and starting at L1.

7. Bar graphs in the Figures will be better to be changed with dot plots. That is a standard these days.

8. On page 12, figure 5, the authors showed the expression of Pirg-1::GFP and Pirg-5::GFP reporters to show the effects of Box C/D snoRNPs on innate immune responses. It will be more convincing if the authors show more targets for further verification, using additional reporters or by performing quantitative RT-PCR.

9. On page 14, the authors mentioned that RNAi targeting box C/D snoRNP components reduced the survival of worms in liquid-based pathogen killing assay, whereas the knockdown increased survival of worms in slow killing assay on agar plates. The authors need to elaborate the conclusion and add discussion for these seemingly contradicting data. The authors also need to change the subheading of this paragraph because it is confusing with respect to the main point of this paper mentioned in the Abstract.

10. In Discussion section, the authors suggested a hypothesis that modification on rRNA by box C/D snoRNP may facilitate translation of specific sub-population of transcripts. A previous paper (Liberman et al., 2020, Science advances; DOI: 10.1126/sciadv.aaz4370) reported that ribosome methylation can facilitate selective translation. I suggest the authors cite this paper in the Discussion section to support the hypothesis.

11. As they mentioned in the Introduction, many papers including Pellegrino et al. 2014, Nature; Kirienko et al. 2015, PNAS; Jeong et al. 2017, EMBO J; Deng et al. 2019, PNAS; Campos et al., 2021 EMBO Rep. have shown that mitochondrial surveillance pathways such as UPRMT activates innate immune responses in C. elegans. They need to cite missing literature and discuss the effects of snoRNPs on protective roles of mitochondrial surveillance systems upon pathogen exposure.

12. For describing Figure 1E, Tiku et al. 2017 needs to be cited and mentioned, because they reported that fib-1 RNAi extends lifespan.

13. Please unify the units (e.g. use ‘hours’ instead of ‘h’ in the method section).

14. Please explain each abbreviation in Figure legends and Methods (on page 22, e.g. SK).

Reviewer #3: This works examines the very interesting but complex interactions among various mitochondrial surveillance pathways including mtUPR, MAPK, and ethanol and stress response network (ESRE) in C. elegans. Here, the authors seek to unravel the transcriptional regulation of the ESRE network and understand the various interacting pathways. Through biochemical mass spec screens, the authors find surprisingly that components of Box C/D SnoRNA methylation complex physically associate with an 11 bp DNA motif comprising the ESRE element. This is unexpected because Box C/D SnoRNA complex is mainly involved in the processing rRNA. Consistent with a regulatory role, knockdown of these components dampens the induction of an ESRE element reporter, and correspondingly the mtUPR reporter hsp-6, but not the mito mapk reporter tbb-6, suggesting specific interaction. In line with specificity, knockdown of the HACA SnoRNA pseudouridylation complex has no such effects. Nor is it simply due to reduced translation, since kd of translational regulators does not inhibit ESRE induction. Further the authors find that knockdown of C/D SNO complex triggers expression of innate immune signaling genes in a manner dependent on the atf-7 transcription factor. Corresponding, worms are protected in slow killing, but not fast killing assays of P. aeruginosa.

This paper represents a conceptual advance as it suggests potential tradeoffs between mitochondrial surveillance and innate immune response, but there are several technical deficiencies that need to be addressed.

1. There is little mechanistic insight into how fibrillarin and nol-56 regulates the ESRE site either directly or indirectly, though the authors remark extensively on this in the discussion. Are the authors suggesting that fib-1 or nol-56 directly bind directly to the ESRE element and act in transcriptional regulation? Or is this a non-specific effect of an abundant protein binding to nucleic acid? To address this, they should also compare wt and mutated ESRE site with worm extracts. Notably, the gel shift in Figure 1 is not publication quality.

2. Reporter constructs can be misleading, particularly if they are multicopy arrays. Can the authors confirm that kd of Box C/D components regulates expression at the level of mRNA of a number of ESRE element containing inducible genes?

3. It has been previously shown that knockdown of nucleolar proteins (fib-1, and nol-6) result in enhanced pathogen resistance, so the connection innate immunity to nucleolar function aspect is not entirely novel. However, extending these observations to the whole C/D SnoRNA complex is interesting, and appears specific since similar physiology is not seen with HACA SnoRNA and pseudouridylation. The question is whether this contrast reflects a difference in the degree of RNAi knockdown, however, and thus levels of transcripts should be quantitated by RT-PCR.

4. Opposite to Box C/D knockdown, the authors nicely show that kd of a number of translational regulators still support activation of the ESRE reporter. Is this also true of small and large subunits of the cytosolic ribosome?

5. A major technical problem could arise from the use of double RNAi. Double RNAi often causes mutual hindrance of the knockdown, and could thus appear as suppression. Empty vector control is not the best one. Instead the authors should use luciferase RNAi, in which the RNAi machinery is induced, but is absent a target, in order to validate their findings in at least a few critical experiments.

6. Figure 3 is confusing and not adequately explained. The authors delete the ESRE element from the ATFS-1 promoter and claim that it has lower levels of hsp-6 expression upon rotenone exposure. Is this a crispr deletion in the chromosome? Judging from the methods it looks like a transgenic insertion. What is it compared to? A wt multicopy array? How can the authors really compare a wt multi copy array to a mutant one without knowing copy number?

7. Overall the paper suggests that snoRNPs are required for activation of mitochondrial stress response ESRE as well as UPRmt, but snoRNP complex was not required for activation of MAPKmt response. Therefore, I don’t understand the model, as it shows UPRmt suppresses MAPKmt (after stress), yet spg-7 RNAi activates both UPRmt as well as MAPKmt, so there should be an activation arrow rather than suppression. Further, there is a clear requirement of snoRNP for UPRmt activation, but it has no effect at all on MAPKmt activation, again not justified by their model.

**Have all data underlying the figures and results presented in the manuscript been provided?**

Reviewer #1: Yes

Reviewer #2: **No: **The list of the identified proteins from their mass spec analysis should be provided as a supplemental table.

Reviewer #3: Yes

PLOS authors have the option to publish the peer review history of their article (what does this mean?). If published, this will include your full peer review and any attached files.

Reviewer #1: No

Reviewer #2: **Yes: **Seung-Jae V. Lee

Reviewer #3: No

---

## [Decision Letter · Decision Letter 1]

14 Feb 2022

Dear Dr Kirienko,

We are pleased to inform you that your manuscript entitled "Box C/D Small Nucleolar Ribonucleoproteins Regulate Mitochondrial Surveillance and Innate Immunity" has been editorially accepted for publication in PLOS Genetics. Congratulations!

Yours sincerely,

Javier E. Irazoqui

Associate Editor

PLOS Genetics

Gregory P. Copenhaver

Editor-in-Chief

PLOS Genetics

Comments from the reviewers (if applicable):

Reviewer's Responses to Questions

**Comments to the Authors:**

Reviewer #1: The authors have sufficiently addressed my concerns. I recommend accepting the current version of the manuscript.

Reviewer #2: The authors addressed all my concerns successfully and the paper has been greatly improved.

**Have all data underlying the figures and results presented in the manuscript been provided?**

Reviewer #1: Yes

Reviewer #2: Yes

PLOS authors have the option to publish the peer review history of their article (what does this mean?). If published, this will include your full peer review and any attached files.

Reviewer #1: No

Reviewer #2: **Yes: **Seung-Jae V. Lee

**Data Deposition**

http://datadryad.org/submit?journalID=pgenetics&manu=PGENETICS-D-21-01267R1

**Press Queries**

---

## [Editor Report · Acceptance letter]

8 Mar 2022

PGENETICS-D-21-01267R1 

Box C/D Small Nucleolar Ribonucleoproteins Regulate Mitochondrial Surveillance and Innate Immunity 

Dear Dr Kirienko, 

We are pleased to inform you that your manuscript entitled "Box C/D Small Nucleolar Ribonucleoproteins Regulate Mitochondrial Surveillance and Innate Immunity" has been formally accepted for publication in PLOS Genetics! Your manuscript is now with our production department and you will be notified of the publication date in due course.

With kind regards,

Katalin Szabo

PLOS Genetics

On behalf of:
